# SUPA: A Lightweight Diagnostic Simulator for Machine Learning in Particle Physics

**Atul Kumar Sinha**
University of Geneva
atul.sinha@unige.ch

**Daniele Paliotta**
University of Geneva
daniele.paliotta@unige.ch

**Bálint Máté**
University of Geneva
balint.mate@unige.ch

**John A. Raine**
University of Geneva, CERN
johnny.raine@cern.ch

**Tobias Golling**
University of Geneva, CERN
tobias.golling@cern.ch

**François Fleuret**
University of Geneva
francois.fleuret@unige.ch

## Abstract

Deep learning methods have gained popularity in high energy physics for fast modeling of particle showers in detectors. Detailed simulation frameworks such as the gold standard GEANT4 are computationally intensive, and current deep generative architectures work on discretized, lower resolution versions of the detailed simulation. The development of models that work at higher spatial resolutions is currently hindered by the complexity of the full simulation data, and by the lack of simpler, more interpretable benchmarks. Our contribution is SUPA, the SUrrogate PArticle propagation simulator, an algorithm and software package for generating data by simulating simplified particle propagation, scattering and shower development in matter. The generation is extremely fast and easy to use compared to GEANT4, but still exhibits the key characteristics and challenges of the detailed simulation. The proposed simulator generates thousands of particle showers per second on a desktop machine, a speed up of up to 6 orders of magnitudes over GEANT4, and stores detailed geometric information about the shower propagation. SUPA provides much greater flexibility for setting initial conditions and defining multiple benchmarks for the development of models. Moreover, interpreting particle showers as point clouds creates a connection to geometric machine learning and provides challenging and fundamentally new datasets for the field.

## 1  Introduction

In order to understand the fundamental building blocks of nature, High Energy Physics (HEP) experiments involve highly energetic particle collisions. These collisions cause particles to decay, and the identification of the resulting decay particles is of key importance to develop and confirm new physics/theories. This is enabled through an electromagnetic or hadronic calorimeter. As particles interact with the calorimeter, they split into multiple other particles, forming a particle shower. The nature of the generated shower depends on the specific material and geometry of the calorimeter, which are carefully selected and driven by physics theory and pratical considerations. The generated shower deposits energy in active layers of the calorimeters, and provides energy and location measurements from the particles produced in the cascade.

37th Conference on Neural Information Processing Systems (NeurIPS 2023) Track on Datasets and Benchmarks.

Precise computer simulation is central to a better understanding of experimental results in HEP. Simulation is especially useful for the task of event reconstruction, where the deposited energy in the calorimeter is used to identify the particle that originated a given shower. The pattern of the deposited energy depends on the specific type of the (intermediate) particles, the initial energy and incidence angle, as well as the specific geometry and shape of the detector material, among other factors. The state-of-the-art for this kind of simulations is GEANT4 [Agostinelli et al., 2003], a Monte Carlo toolkit for modeling the propagation of particles through matter.

While detailed simulation through GEANT4 provides fine-grained shower generation and captures the underlying distributions accurately, this software consists of more than 3.5 million lines of C++, is computationally very expensive, and requires specific domain knowledge to be set up and tuned. This makes it cumbersome and expensive to produce the amount and diversity of data needed to speed up machine learning research on these topics. Table 1 compares GEANT4 with our proposed simulator SUPA.

Our key contributions to ease these issues are:

- We introduce SUPA: a fast, easy to use simulator for simple particle propagation, scattering and shower development in matter (see § 3).

- We structure the simulator in a way that allows to easily change the complexity of the underlying shower development as well as the properties of the showers in terms of multiplicity, energy range, structure (see § 4.1).

- We use SUPA to generate data that highlights the limitations of current machine learning approaches for fast shower simulation (see § 5.2). We experimentally demonstrate that SUPA can be used as a proxy for detailed simulation, while being easy to tune and up to 6 orders of magnitude faster than GEANT4.

## 1.1 Image Representation

Existing approaches for generating particle showers [Paganini et al., 2018, Krause and Shih, 2021] focus on a quantized representation of the calorimeter hits, binning them into discrete pixels, or cells, and setting the pixel intensity to the sum of the energy deposited. The upside of this approach is that, due to the structure of existing calorimeters, the data takes the form of low resolution images and standard computer vision models can be directly applied. However, as we increase the resolution of the quantized representations in order to preserve more information, the resulting images become exceedingly sparse, leading to difficulties in training existing machine learning architectures. Figure 1b, 1c, and 1d show the downsampled image representations at resolutions of 3x, 2x and 1x respectively for the shower shown in Figure 1a. We choose 1x to be the same resolution as used in CaloGAN [Paganini et al., 2018] (i.e. $12 \times 12$ for Layer 1).

It is also important to notice that, when decreasing the resolution of the data and looking at showers, while the signal to model is of lower dimension and less sparse, the underlying structure and relationships between hits are lost. This requires models to implicitly learn equivariances from multiple examples, while also introducing artificial structure arising from the chosen detector geometry. For example, with a shower incident on the surface of a detector, a slight translation in either direction does not change the properties of the shower. However it still has an impact on the resulting datasets, where for small shifts it is not translationally equivariant. Having access to the underlying energy depositions would therefore greatly reduce the dependence on the geometry of the detectors in the models, and allow generative models to learn the underlying structure from the physical process directly.

## 1.2 Point Cloud Representation

Many popular point-cloud datasets [Wu et al., 2015, Chang et al., 2015] in the machine learning community contain shape representations of physical objects, such as points sampled uniformly from the surfaces of chairs, tables, etc. These datasets enjoy many nice properties, e.g. locality and redundancy, that calorimetry datasets do not. Moreover, existing generative models for point clouds rely on these simplifying assumptions and therefore their performance on more diverse datasets is still relatively unexplored. In the case of particle showers in calorimeters, dealing directly with the point cloud representation provides an optimal description of the underlying physics processes. However,

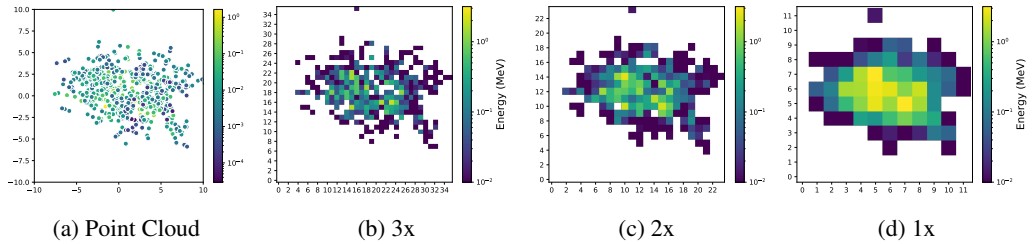

| (a) Point Cloud | (b) 3x | (c) 2x | (d) 1x |

Figure 1: An example shower (single layer) shown at different resolutions

due to the detailed simulation resulting from GEANT4, the number of points per shower represents a huge computational challenge.

A key objective of our synthetic simulator is to bridge this gap by allowing for quicker ground truth data generation for different initial settings, thus providing a way of modulating the complexity of the dataset, leading to more interpretable benchmarks. This would also make it possible to take gradual steps in developing the models and work out which of the many complexities in the GEANT4 datasets are hardest to model.

## 2   Related Work

Shower simulation with GEANT4 requires a complete description of the geometry and material of the detector and simulates all interactions between the initial particle, any subsequently produced particles through decays or emissions, and the detector material. The upside of using GEANT4 for simulation is that it is very accurate and represents the underlying physical phenomena precisely, but it comes at the cost of being computationally very expensive. Moreover, it requires expertise to define the configuration and a multitude of expert hours to set it up for new detector geometries. To reduce the computational resources required to simulate particle physics collisions and their energy depositions in detectors, deep generative models have been applied to generate particle showers. Most of these models [Abhishek et al., 2021, Krause and Shih, 2021, Paganini et al., 2018] employ the dataset first introduced in CaloGAN [Paganini et al., 2018] which is based on a simplified calorimeter inspired by the ATLAS Liquid Argon (LAr) electromagnetic calorimeter. [1]

### 2.1   Datasets and calorimeter structure

For the CaloGAN dataset, the calorimeter is cubic in shape with each dimension being 480 mm and no material in front of it. The volume is divided into three layers along the radial ($z$) direction with varying thicknesses of 90 mm, 347 mm, and 43 mm, respectively. These layers are further segmented into discrete cells, with different sizes for each layer: 160 mm $\times$ 5 mm (first layer), 40 mm $\times$ 40 mm (second layer) and 40 mm $\times$ 80 mm (third layer). Each layer can be represented as a single-channel two-dimensional image with pixel intensities representing the energy deposited in the region. The final read-out has the resolution of $3 \times 96$, $12 \times 12$, and $12 \times 6$. For this dataset, three different particle types were considered, namely, positrons, charged pions and photons. Further, the particles were configured to be incident perpendicular to the calorimeter with initial energies uniformly distributed in the range between 1 GeV and 100 GeV.

Several other calorimeter geometries and configurations are employed across the literature. Erdmann et al. [2019] consider a calorimeter configuration motivated by the CMS High Granularity Calorimeter (HGCAL) prototype, while Belayneh et al. [2020] study a calorimeter based on the geometric layout of the proposed Linear Collider Detector (LCD) for the CLIC accelerator. Buhmann et al. [2021b,a] investigate the prototype calorimeter for the International Large Detector (ILD) which is one of the two proposed detector concepts for the International Linear Collider (ILC). The Fast Calorimeter Simulation Challenge 2022 (CaloChallenge2022) Faucci Giannelli et al. [2022c,a,b] was proposed to spur the development and benchmarking of fast and high-fidelity calorimeter shower generation using deep learning methods. The challenge proposed three datasets, ranging in difficulty from easy

---
[1]The ATLAS calorimeter has e.g. a more complex geometry, with the cells having accordion shaped electrodes to maximise the active volume

to medium to hard. the difficulty is determined by the dimensionality of the calorimeter showers i.e. number of layers and number of voxels in each layer. In a different application setting, Erdmann et al. [2018] model a calorimeter response for cosmic ray-induced air showers in the Earth's atmosphere, producing signals in ground based detector stations. In contrast to the CaloGAN calorimeter, it has a single readout layer with $9 \times 9$ cells, each cell corresponds to a detector unit placed with a spacing of $1500\ m$. Although many different detector technologies are considered in the various models, the common aspect between the models and the corresponding datasets is the projection of the spatial signal onto discretized cells in 2D planes or a generalized 3D volume. The projection is done either directly while designing the calorimeter geometry, or as a post processing step. Further, the number of cells and thus the resolution of the final read-out is usually kept small. This is done either to simplify the data for training the models, or in order to speed up the simulations.

Due to the different geometries used for generating the datasets, it is not a straightforward task to compare the performance of the various models. As the developments of each shower are completely dependent on the detailed model of the detector in GEANT4, although the same initial particles can be simulated, it is very difficult to draw parallels between two different datasets, as it is not possible simply to change the representation of the data from one model to look like that of another.

## 2.2 Deep Generative Models

The applications of deep generative modeling to calorimeter simulation have so far almost entirely focused on Generative Adversarial Networks (GANs, Goodfellow et al. 2014) and Normalizing Flows [Rezende and Mohamed, 2016]. CaloGAN [Paganini et al., 2018] was the first application of GANs to a longitudinally segmented calorimeter. The approach is based on LAGAN [de Oliveira et al., 2017], a DCGAN [Radford et al., 2016]-like architecture, that is able to synthetize the shower images. The generator outputs a gray-scale image for each layer in the calorimeter, with each output pixel representing the energy pattern collected at that location. The architecture is also complemented with an auxiliary classifier tasked with reconstructing the initial energy $E$.

Krause and Shih [2021] improve on CaloGAN with CaloFlow, a normalizing flow architecture to generate shower images. CaloFlow is able to generate better samples than previous approaches based on GANs and VAEs, while also providing more stable training. Inductive CaloFlow (iCaloFlow) Buckley et al. [2023] extends the CaloFlow model to higher granularity detector geometries. It utilizes a teacher-student distillation to increase sampling speed without loss of expressivity.

Previous work by ATLAS collaboration [2018] proposes to use a Variational Autoencoder on the flattened images [Kingma and Welling, 2014] . Both the encoder and decoder consist of an MLP conditioned on the energy of the initial particle. In addition to the reconstruction and KL-terms, the authors also optimize for the overall energy deposit in both in the individual layers and in the overall system.

Buhmann et al. [2023] proposes a diffusion-based generative model representing one of the initial methods devised for effectively processing the point cloud representation of calorimeter showers.

## 3 Propagation Model

SUPA generates data that imitates the propagation and splitting/scattering of point particles. To generate an event, a point particle is initialized at the origin of the 3D space with an initial energy [2] value and a velocity parallel to the z-axis. We then define a sequence of 2D planes orthogonal to the $z-$axis at $\{z_0, ..., z_N\}$ that we call "slices" (or "sub-layers", interchangeably). We assume that the slices are evenly spaced along the $z$-axis, with a gap of $\Delta z$ between consecutive slices. As the initial particle reaches the first predefined $z$-slice at $z_0$, it has 3 options (see Figure 2): stop and deposit its energy with *stopping probability* $p_{\text{stop}}$, split into two with *splitting probability* $p_{\text{split}}$, or simply continue moving with the same velocity with *pass-through probability* $p_{\text{pass}} = 1 - (p_{\text{stop}} + p_{\text{split}})$. Then:

- If the particle splits, two particles are generated at the splitting location with energy and velocity features that ensure momentum conservation, i.e., that the sum of momentum

---

[2]A continuous attribute obeying simple conservation laws at each splitting, which can be interpreted as an equivalent for the energy of the particles in a shower. As such, we refer to this property as the energy.

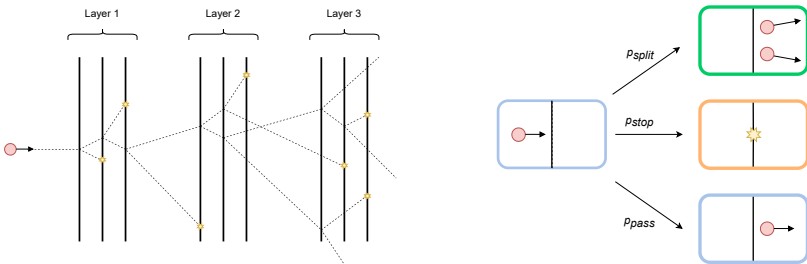

Figure 2: Schematic summary of a propagation event generated by SUPA. (left) Splitting events can happen at the black vertical lines, called *slices*. See § 3 for a detailed explanation of the process. (right) Each time a particle crosses a pre-defined "slice", it either splits in two, stops and deposits energy, or passes through unperturbed.

vectors of the new particles is equal to the decaying one. The original particle is removed from the propagation model and the two new ones continue propagating with their respective velocities. The *splitting angle* $\theta$ and *deviation angle* $\epsilon$ are sampled from distributions $p_\theta(.)$ and $p_\epsilon(.)$, respectively.

- If the particle stops, its energy is deposited and the energy value and the location of the hit are recorded.

- If the particle passes through, it simply continues with its original velocity.

See Algorithm 1 for the pseudocode of this process. We can think of the process as a detector layer being positioned at the $z_0$-slice recording energy deposits. After the slice at $z_0$, the model either contains 0,1 or 2 particles still propagating. As they reach the additional slices at $\{z_1, ..., z_N\}$ the above procedure is repeated: whenever a particle is at a $z$-slice, it either passes, deposits its energy or decays.

After all particles crossed the last slice at $z_N$ we collect all energy deposits (energy value and 3D location). Instead of building a dataset from the exact 3D location of these deposits, we partition the slices into subsets we call *layers*. We refer to the partitioning scheme as *layer configuration*.

We then build an event from the energy deposit $E$ and the $(x, y)$-coordinate of all hits for all slices within a given layer $L_i$, effectively projecting the z-axis into a single point. An event then consists of $N$ 2D point clouds, one for each layer, with a single scalar feature per point. Intuitively, the slices $z_i$ are the positions where the particle can split and stop (granularity of the simulation), and the layers $L_i$ correspond to the detector layers, where the readout happens. Figure 2 contains a schematic summary for the propagation process.

Having complete control over the algorithm, we can increase/decrease the complexity of the generated data by adjusting the distribution of the properties of the initial particle, the number and location of the slices $z_i$, the probabilities $p_{\text{stop}}, p_{\text{pass}}, p_{\text{split}}$ and the distribution of the properties of the children after a splitting event. This is especially useful to debug and understand the behaviour of machine learning models at different levels for complexity.

Table 1: Comparison of GEANT4 and the proposed model (SUPA)

|  | GEANT4 | SUPA |
|---|---|---|
| codebase | 3.6m lines in C++ | few hundred lines in Python |
| target usage | detailed physics simulations | fast data generation |
| speed | 1772 ms/shower[3] | 0.1-100 ms/shower[4] |

## Algorithm 1 SUPA - Particle Splitting

**Parameters:** Stopping probability: $p_{stop}$
Splitting probability: $p_{split}$
Layer configuration :
$\qquad L_1 = \{z_0, ..., z_{i_1}\},$
$\qquad L_2 = \{z_{i_1+1}, ..., z_{i_2}\},$
$\qquad ...$
$\qquad L_n = \{z_{i_{n-1}+1}, ..., z_N\},$
Slice Gap: $\Delta z$
Splitting angle: $p_\theta(.)$
Deviation angle: $p_\epsilon(.)$

**Input:** Tree $\mathcal{T}$, Position $\mathbf{p} = (\Delta\eta, \Delta\phi)$,
Energy $E$, Layer $l$

**Output:** a collection of $\{\mathbf{p}_i, E_i\}^N$
of generated particles

1: $(\mathbf{p}, E, l) \leftarrow$ **Input**, $N \leftarrow p(\delta(n=2))$

2: $\theta \sim p_\theta(.), \epsilon \sim p_\epsilon(.), \alpha \sim \mathcal{U}(0,1)$
3: **if** $\alpha < p_{stop}[l]$ **then**
4: $\quad$ Active $\leftarrow$ False
5: **else if** $p_{stop}[l] < \alpha < p_{stop}[l] + p_{split}$ **then**
6: $\quad \omega \leftarrow \mathcal{U}(0, 2\pi)$
7: $\quad \mathbf{p_0} \leftarrow \mathbf{p} + \Delta z * (\sin(\theta/2 + \epsilon), \cos(\theta/2 + \epsilon))$
8: $\quad \mathbf{p_1} \leftarrow \mathbf{p} - \Delta z * (\sin(\theta/2 + \epsilon), \cos(\theta/2 + \epsilon))$
9: $\quad E_{\{0,1\}} \leftarrow E * (\theta/2 \pm \epsilon)/\theta$
10: $\quad \mathcal{T}.add(\mathbf{p_0}, l+1, E_0), \mathcal{T}.add(\mathbf{p_1}, l+1, E_1)$
11: **else**
12: $\quad \mathcal{T}.add(\mathbf{p}, l+1, E)$
13: **end if**
14: Return $\mathcal{T}$

**Notation.** We denote a single shower as $\mathbf{x} = \{\mathbf{x}_{L_1}, \mathbf{x}_{L_2}, \ldots, \mathbf{x}_{L_N}\}$, where $\mathbf{x}_{L_i}$ denotes the $i^{\text{th}}$ layer and $N$ is the total number of layers considered. For the point cloud representation, $\mathbf{x}_{L_i}$ can be further expanded as $\mathbf{x}_{L_i} = \{\mathbf{x}_1^i, \mathbf{x}_2^i, \ldots, \mathbf{x}_{N_i}^i\}$, where $N_i$ denotes the total number of points in layer $i$ of the shower (can vary across showers). Further, each point is a vector $\mathbf{x}_j^i = [\eta_j^i, \phi_j^i, E_j^i]$, where $\eta$ and $\phi$ are the coordinates and $E$ denotes the energy.

## 4 Using SUPA

SUPA provides various meta-parameters which affect the data generation and various characteristics of the generated data. This allows to generate shower data with varying levels of complexity and thus enables incremental benchmarking. For instance, $p_{split}$ affects the number of emerging secondary particles and thus implicitly the number of hits (or points), while $p_{stop}$ controls the position (in depth) of the observed hits. $\theta$ controls the overall spread of the hits in the lateral plane and $\alpha$ controls the dynamic range of the observed energy values. The different configurations for these parameters is analogous to different detector geometry/material or perhaps a different particle.

### 4.1 SUPA Datasets

We propose five standard configurations and the resulting datasets SUPAv1-5 for our analysis and benchmarking, summarized in Table 2. All configurations have the same initial energy of 65 GeV, initial angle of $\pi/2$ and initial impact position $(0, 0)$, i.e. particles are incident perpendicular and at the center of the calorimeter. Section A.2 in the appendix gives more details and also shows the interpretation of these variations with respect to different summary variables. SUPAv1 is deterministic

Table 2: Summary of different datasets generated with SUPA

| Dataset | $\theta$ | $\alpha$ | $p_{split}, p_{stop}, p_{pass}$ | # Points | Layer Configuration |
|---------|----------|----------|--------------------------------|----------|---------------------|
| SUPAv1 | $\frac{\pi}{24}$ | 0 | see Fig. 5a | 128 | L0 : [5, 10] |
| SUPAv2 | $\frac{\pi}{24}$ | 0 | see Fig. 5b | [1, 91] | L0 : [7, 20] |
| SUPAv3 | $\mathcal{U}(\frac{\pi}{32}, \frac{\pi}{16})$ | $\mathcal{U}(\frac{-\theta}{4}, \frac{\theta}{4})$ | see Fig. 5b | [1, 91] | L0 : [7, 20] |
| SUPAv4 | $\mathcal{U}(\frac{\pi}{32}, \frac{\pi}{16})$ | $\mathcal{U}(\frac{-\theta}{4}, \frac{\theta}{4})$ | see Fig. 5b | [1, 84] | L0 : [7, 12] |
| SUPAv5 | $\mathcal{U}(\frac{\pi}{32}, \frac{\pi}{16})$ | $\mathcal{U}(\frac{-\theta}{4}, \frac{\theta}{4})$ | see Fig. 5c | [5, 280] | L0 : [7, 12], L1 : [13, 28] |

---

[3]The exact simulation time depends on the incident particle and its kinematic properties, as well as the detailed composition of the detector geometry in the GEANT4 model. These numbers are taken for the CaloGAN geometry introduced in Paganini et al. [2018].

[4]Reflects the time to generate directly the point cloud representation, which is more granular than the corresponding GEANT4 generation

with respect to splitting and stopping probabilities as well as in $\theta$ and $\alpha$. It has the same number of points across all events and all the hits have the same energy value. SUPAv2-4 have the same configuration for splitting and stopping probabilities (see Sec. A.2.1), while SUPAv2 is deterministic in $\theta$ and $\alpha$. SUPAv3 is naturally more spread out than SUPAv2 (see Fig. 6b vs. Fig. 6c, and also distributions for $\eta$, $\phi$, $r$ in Fig. 7, and $\langle\eta\rangle$, $\langle\phi\rangle$, $\langle r\rangle$ in Fig. 8). SUPAv4 is similar to SUPAv3, except for the layer configuration. SUPAv4 only has a subset of the sub-layers by dropping the last 8 sub-layers from SUPAv3. While SUPAv3 captures the total initial energy in Layer 0, SUPAv4 has some energy leakage which is apparent from the distribution of layer energy $\bar{E}$ in Fig. 10f. Another interesting structure in shower data is the dynamic range of the energy values in a given shower. A high dynamic range of energy makes learning generative models difficult [Krause and Shih, 2021]. With SUPA, we can control the dynamic range via the parameter $\alpha$. SUPAv1 and SUPAv2 have fixed $\alpha(=0)$, thus all the splits are symmetric leading to a low dynamic range (see Fig. 10d for histogram of variance in energy values). The other factor which affects the dynamic range is the layer configuration, more number of slices in a layer would lead to more dynamic range. SUPAv5 has multiple layers and has a higher $p_{split}$ in the initial slices, thus it is more complex in terms of multiplicity or total number of hits. Please refer to the appendix Sec. A.2 for more details and other canonical SUPA datasets.

## 4.2 Extensions

SUPA is very flexible and can be easily extended. The slices can be further divided into different regions each with their own meta-parameters providing more granular control over the scattering and splitting process. Another variation requires having some slices that are not merged into layers to mimic dead material. These extensions can possibly bring it closer to more realistic detectors, and lead to unique structures in the dataset and thus present challenges for model development and evaluation. It is also possible to condition the data generation on various attributes, for example: initial impact angle, impact energy, impact position, etc. This encourages interesting studies and benchmarks regarding extrapolation and generalization capabilities of generative models for unseen conditioning variables during training.

## 5 Evaluation and Analysis

The evaluation of generative models for high dimensional data is a non-trivial problem. While reconstruction losses and/or data likelihood are available for some model architectures (e.g. auto-encoders or flows), judging the sample quality is still difficult. Moreover, the average log-likelihood is difficult to evaluate or even approximate for many interesting models. For instance, in computer vision, evaluating generative models for images is done via proxy measures such as the Frechet Inception Distances (FID, Heusel et al., 2017). Point cloud generative models are evaluated [Yang et al., 2019] using metrics such as Minimum matching distance (MMD), Coverage (COV) and 1-Nearest Neighbour Accuracy (1-NNA), where similarity between a set of reference point clouds and generated point clouds are measured using a distance metric such as the Chamfer distance (CD) or Earth mover's distance (EMD) based on optimal matching.

### 5.1 Shower Shape Variables

For calorimeter simulation, we cannot judge performance using individual examples due to the stochastic nature of shower development. The standard practice to estimate the quality of generative models for calorimetry data (shower simulations) is to use histograms of different metrics called *shower shape variables* [Paganini et al., 2018] in addition to images of calorimeter showers. These shower shape variables are domain-specific and physically motivated, and matching densities over these marginals is indicative of matching salient aspects of the data. These marginals capture various aspects of how energy is distributed within individual layers and across different layers.

We extend the shower shape variables for the point clouds representation of showers. A summary of all the shower shape variables considered are present in the appendix.

**Point level marginals**. Marginals of each point feature by considering the set all the points from all the point clouds together.

**Feature means** $\langle\eta_i\rangle, \langle\phi_i\rangle, \langle r_i\rangle, \langle E_i\rangle$. Mean of each feature.

$$\langle\eta_i\rangle = \frac{\sum_j \eta_j^i}{\sum_j 1}, \ \langle\phi_i\rangle = \frac{\sum_j \phi_j^i}{\sum_j 1}, \ \langle r_i\rangle = \frac{\sum_j r_j^i}{\sum_j 1} \ \langle E_i\rangle = \frac{\sum_j E_j^i}{\sum_j 1}$$

where $r_j^i = \sqrt{(\eta_j^i)^2 + (\phi_j^i)^2}$ denotes the distance of the point in the lateral plane from the center.

**Feature variances** $\sigma_{\langle\eta_i\rangle}, \sigma_{\langle\phi_i\rangle}, \sigma_{\langle r_i\rangle}, \sigma_{\langle E_i\rangle}$. Variance of each feature. $\sigma_{\langle\eta_i\rangle} = \sqrt{\frac{\sum_j \eta_j^{i\,2}}{\sum_j 1} - \langle\eta_i\rangle^2}$

**Layer Energy** $\bar{E}_i$. Denotes the total energy deposited in layer $i$ of the shower. $\bar{E}_i = \sum_{j\in N_i} E_j^i$.

**Total Energy** $E_{\text{tot}}$. Total energy across all layers of the shower. $E_{\text{tot}} = \sum_{i\leq N} \bar{E}_i$.

**Layer Centroids** $\langle\eta_i\rangle_E, \langle\phi_i\rangle_E, \langle r_i\rangle_E$. Energy weighted mean of the features ($\eta$, $\phi$, or $r$).

$$\langle\eta_i\rangle = \frac{\sum_j E_j^i \eta_j^i}{E_i}, \ \langle\phi_i\rangle = \frac{\sum_j E_j^i \phi_j^i}{E_i}, \ \langle r_i\rangle = \frac{\sum_j E_j^i r_j^i}{E_i}$$

The layer centroids can be interpreted as the center of energy in the lateral plane in respective dimensions.

**Layer Lateral Width** $\sigma_{\langle\eta_i\rangle_E}, \sigma_{\langle\phi_i\rangle_E}, \sigma_{\langle r_i\rangle_E}$. Denotes the standard deviation of the layer centroids.

$$\sigma_{\langle\eta_i\rangle_E} = \sqrt{\frac{\sum_j E_j^i (\eta_j^i)^2}{E_i} - \langle\eta_i\rangle_E^2}$$

The layer lateral widths can be interpreted as the spread around the center of energy in the lateral plane in respective dimensions. We drop the layer notation $i$ from the above metrics when working with a single layer for brevity.

## 5.2 Performance Analysis of Generative Models

In order to summarise the discrepancy into a scalar value, we consider the Wasserstein-1 distance between the histograms of the ground truth and generated sample's marginals[5]. Further, we compute the **mean discrepancy** over different groups of marginals to summarize the performance of models.

We train point cloud generative models, PointFlow [Yang et al., 2019], SetVAE [Kim et al., 2021], and a transformer-based flow model (which we call *Transflowmer*, see Sec. A.3), on SUPA data. Table 3 summarizes the performance of different models across different variations of SUPA datasets. Pointflow performs better than SetVAE for all marginals not involving energy feature such as point level marginals, feature means and variances for location parameters, etc., and worse for marginals which depend on energy such as, layer centroids, and layer energy, etc. (with the exception of layer lateral widths and feature variance $\sigma_{\langle E\rangle}$, which are both second order moments), across all datasets. Roughly speaking, PointFlow struggles at modeling the energy feature while it is competitive for the location features, and SetVAE behaves opposite. It is interesting to observe that Transflowmer is either the best performing or very close to best performing model (either SetVAE or PointFlow) across all datasets and marginals, indicating relatively greater flexiblity at modeling a variety of features. The instances with relatively high values in Table 3 (feature means for SUPAv1, and layer energy for SUPAv2 and SUPAv3) are where the marginals had a very narrow spread, indicating the difficulty in modeling discrete distributions. The models also have the tendency to predict points with very high energy values unseen during training (see Fig. 24a). We provide more details in the Appendix Section A.4 on the training, results and analysis, along with plots of various shower shape variables to visually compare the goodness of fit.

These observations highlight the need for adapting point cloud generative models such that they are able to model points with special features (different from location parameters), such as physical features like energy, momentum, etc. Although the location parameters are correlated with these special features, the overall structure of this correlation can be very different from domain to domain and also from how they are correlated with other location features.

---

[5]For computing the Wasserstein distance, we normalize the shower shape variables for the ground truth and the generated sample according to the mean and standard deviation of the ground truth.

Table 3: Performance benchmarks across different datasets with SetVAE, PointFlow and Trans-flowmer. The distance metric is Wasserstein-1. The reported numbers are averages over a group of marginals as indicated in the top row. Lower numbers are better. See § 5.1 for more details on the metrics.

| Dataset | Point Features : $\eta, \phi, r$ | | | Point Feature : $E$ | | | $\langle\eta_i\rangle, \langle\phi_i\rangle, \langle r_i\rangle$ | | |
|---|---|---|---|---|---|---|---|---|---|
| | SV | PF | TF | SV | PF | TF | SV | PF | TF |
| SUPAv1 | 0.044 | 0.037 | **0.028** | 0.001 | **0.000** | **0.000** | 16.254 | 21.921 | **5.789** |
| SUPAv2 | 0.290 | 0.167 | **0.029** | 0.253 | 0.462 | **0.145** | 0.517 | 0.256 | **0.091** |
| SUPAv3 | 0.510 | 0.130 | **0.044** | 0.304 | 0.484 | **0.048** | 0.769 | **0.075** | 0.105 |
| SUPAv4 | 0.269 | 0.122 | **0.029** | 0.111 | 0.459 | **0.006** | 0.460 | 0.094 | **0.073** |
| SUPAv5 | 0.166 | **0.028** | 0.042 | **0.078** | 0.090 | 0.151 | 0.592 | **0.031** | 0.189 |

| Dataset | $\sigma_{\langle\eta_i\rangle}, \sigma_{\langle\phi_i\rangle}, \sigma_{\langle r_i\rangle}$ | | | $\langle E\rangle$ | | | $\sigma_{\langle E\rangle}$ | | |
|---|---|---|---|---|---|---|---|---|---|
| | SV | PF | TF | SV | PF | TF | SV | PF | TF |
| SUPAv1 | 0.513 | 0.585 | **0.359** | 0.001 | **0.000** | **0.000** | **0.000** | **0.000** | **0.000** |
| SUPAv2 | 0.648 | 0.154 | **0.130** | 0.302 | **0.077** | 0.087 | 0.513 | 0.177 | **0.165** |
| SUPAv3 | 1.114 | **0.109** | 0.126 | 0.320 | **0.064** | 0.071 | 0.500 | 0.152 | **0.144** |
| SUPAv4 | 0.634 | 0.092 | **0.051** | 0.263 | 0.047 | **0.022** | 0.355 | 0.156 | **0.038** |
| SUPAv5 | 0.799 | **0.040** | 0.223 | 0.251 | **0.059** | 0.414 | 0.377 | **0.047** | 0.421 |

| Dataset | $\langle\eta_i\rangle_E, \langle\phi_i\rangle_E, \langle r_i\rangle_E$ | | | $\sigma_{\langle\eta_i\rangle_E}, \sigma_{\langle\phi_i\rangle_E}, \sigma_{\langle r_i\rangle_E}$ | | | $\bar{E}$ | | |
|---|---|---|---|---|---|---|---|---|---|
| | SV | PF | TF | SV | PF | TF | SV | PF | TF |
| SUPAv1 | 16.244 | 21.921 | **5.766** | 0.517 | 0.591 | **0.379** | 0.101 | **0.000** | 0.039 |
| SUPAv2 | 1.336 | 1.933 | **0.137** | 0.779 | **0.134** | 0.190 | 23.387 | 69.814 | **10.468** |
| SUPAv3 | 1.365 | 1.627 | **0.217** | 1.226 | **0.147** | 0.151 | 36.116 | 79.062 | **5.916** |
| SUPAv4 | 1.369 | 1.346 | **0.083** | 0.645 | 0.169 | **0.067** | 3.997 | 12.895 | **0.659** |
| SUPAv5 | 2.373 | 1.615 | **0.463** | 0.740 | **0.058** | 0.317 | **6.132** | 16.742 | 9.273 |

## 5.3 SUPA as a benchmark

We train the models of § 2.2 on both the events generated by GEANT4 and SUPA. For these studies, we generated another version of the dataset with SUPA and downsample the point clouds to grid representation such that it is similar to the CALOGAN dataset (see § A.5 for details on data generation). The training loss of the flow model is a log-likelihood which meaningfully represents the quality of the captured distribution. Figure 3 displays the correlation of the log-likelihood values of different variations of the CaloFlow architecture trained both on GEANT4 and SUPA-generated events. Figure 4 shows the scatter plot of the mean discrepancy (across all marginals) obtained by different models on GEANT4 and SUPA-generated events. We show more plots in the Appendix § A.5.

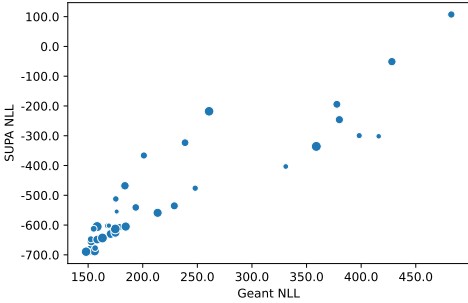

Figure 3: Scatter plot for loss over different versions of CaloFlow models. Each point corresponds to an architecture, its $x$ and $y$ coordinates are the negative log-likelihoods on data generated by GEANT4 and SUPA, respectively. Point size reflects the capacity of the model (number of layers, learnable parameters, etc.).

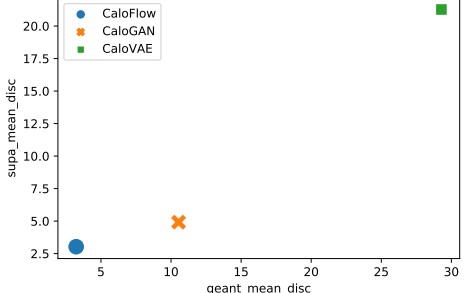

Figure 4: Scatter Plot for mean discrepancy (§ 5) over different models. Lower numbers are better. Performance of models are consistent over both datasets, a better model on SUPA implies a better model on GEANT4 and vice versa.

We can observe that all these correlation plots are roughly monotonic, showing that benchmarking on SUPA provides a good estimate of relative performance on the detailed GEANT4.

## 6 Conclusion

We introduced SUPA, a lightweight pseudo-particle simulator that is inspired by the physics governed development of particle showers and qualitatively resembles the data generated by the gold standard GEANT4. By allowing to easily change the underlying parameters of the propagation model, we introduce the freedom to define new benchmark datasets of varying complexity with a fidelity and simplicity not available using the GEANT4 toolkit.

We believe that the ease of use, flexibility and speed of this simulator could be a MNIST moment for the field, allowing the rapid exploration of models in various regime of functioning as discussed in § 4.1. We also highlight the gaps in the ability of existing point cloud generative models on calorimetry data in § 5.2 and highlight the need for more powerful point cloud generative models.

Additionally, we showed in § 5.3 and § A.5.1 that with grid representation of the shower data, performance of deep generative models estimated on data produced with this simulator is a good proxy for their performance on the standard data used historically in machine learning for particle physics. We highlight flexibility of SUPA in § A.5.2, allowing us to train/evaluate at different resolutions.

Moreover, by giving easy access to the fully recorded event, we open the way to applying methods from point clouds and geometric deep learning research, with the goal of possibly devising end-to-end architectures able to reconstruct a full event from the energy deposits by solving the complete inverse problem, something currently impossible with data from GEANT4.

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

# A Appendix

## A.1 Shower shape variables

We extend the list of shower shape variables described in Sec. 5.1 :

**Layer Energy Fraction** $f_i$. Fraction of the total energy deposited in layer $i$ of the shower. $f_i = \bar{E}_i/E_{\text{tot}}$.

**Energy Ratio** $E_{\text{ratio},i}$. Ratio of the difference between highest and second highest energy intensity point or cell in layer $i$ and their difference. $E_{\text{ratio},i} = \frac{E_{[1]}^i - E_{[2]}^i}{E_{[1]}^i + E_{[2]}^i}$.

**Depth** $d$. Deepest layer in the shower with non-zero energy deposit. $d = \max_i\{i : \max_j(E_j^i) > 0\}$.

**Layer/Depth Weighted Total Energy** $l_d$. Sum of the layer energies weighted by the layer number. $l_d = \sum_{i \leq N} i \cdot \bar{E}_i$.

**Shower Depth** $s_d$. Depth weighted total energy normalized by the total energy in the shower. $s_d = l_d/E_{\text{tot}}$.

**Shower Depth Width** $\sigma_{s_d}$. Standard deviation of $s_d$ in units of layer number.

$$\sigma_{s_d} = \sqrt{\frac{\sum_{i=0}^2 i^2 \cdot \bar{E}_i}{E_{\text{tot}}} - \left(\frac{\sum_{i=0}^2 i \cdot \bar{E}_i}{E_{\text{tot}}}\right)^2}$$

**Brightest Voxels** $E_k\_\text{brightest\_layer}_i$. The $k^{\text{th}}$ brightest voxel in layer $i$ normalized by the total layer energy. $E_k\_\text{brightest\_layer}_i = E_{[k]}^i/E_i$.

**Layer Sparsity**. The ratio of the number of cells with non-zero energy to the total number of cells in layer $i$. This is only valid for the image representation of showers.

## A.2 Details on different variations of SUPA datasets

### A.2.1 Parameters

Fig. 5 shows the remaining parameters used for generating SUPA variations (see Table 2 for details on other parameters). SUPAv1 is most deterministic as particles always split in the first six sub-layers with no deposits ($p_{split} = 1$ and $p_{stop} = 0$ for all sub-layers < 7), further since $p_{stop} = 1$ at sub-layer 7, all the particles get deposited. Thus each event/example in SUPAv1 has exactly $128(= 2^7)$ points. Further, since $\alpha$ is fixed to 0, all splits are symmetric and energy is always halved at each split, thus all deposits have the same energy value. SUPAv5 has higher $p_{split}$ in the initial sub-layers ($< 7$) than SUPAv2-4, while $p_{stop}$ is the same for all of them, thus SUPAv5 has more number of hits/points than SUPAv2-4 in the respective sub-layers or layers.

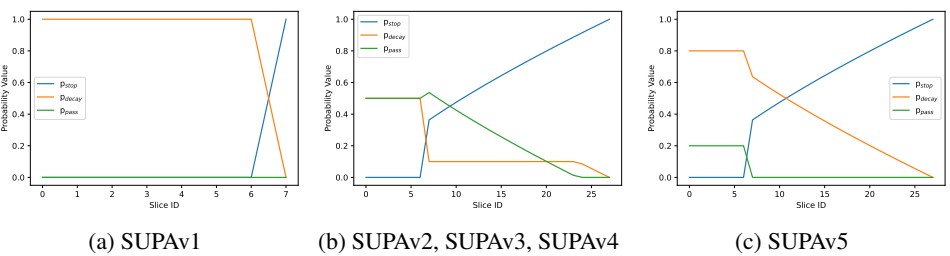

|  (a) SUPAv1 | (b) SUPAv2, SUPAv3, SUPAv4 | (c) SUPAv5 |

Figure 5: Parameters $p_{split}, p_{stop}, p_{pass}$ for SUPA variations

### A.2.2 Shower Shape Variables

Fig. 6 shows the average events for different variations of SUPA datasets and Figs. Fig. 7 - 12 shows the histograms of the various shower shape variables for all SUPA datasets.

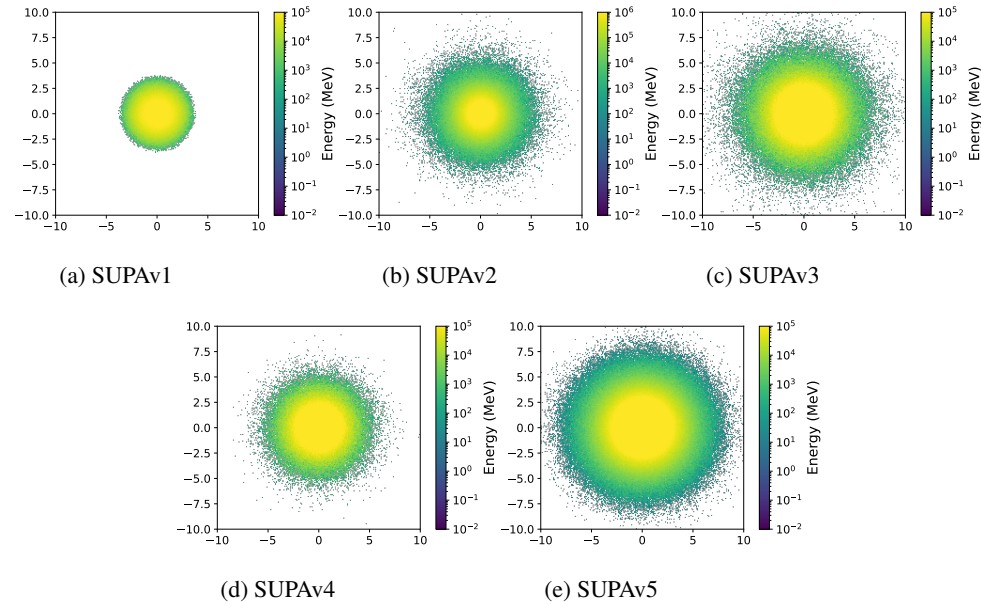

(a) SUPAv1      (b) SUPAv2      (c) SUPAv3

(d) SUPAv4      (e) SUPAv5

Figure 6: Average event representation for different variations of SUPA datasets

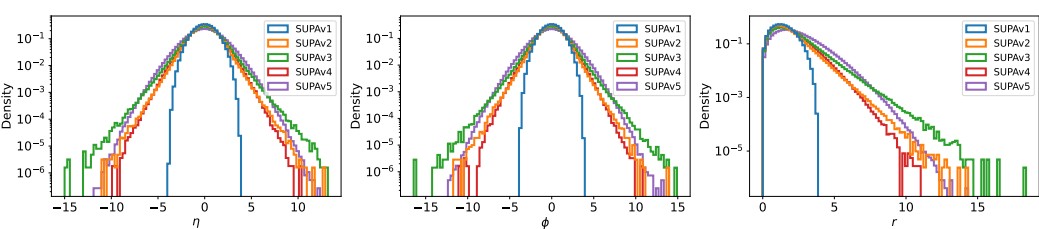

Figure 7: Histograms of point level distributions

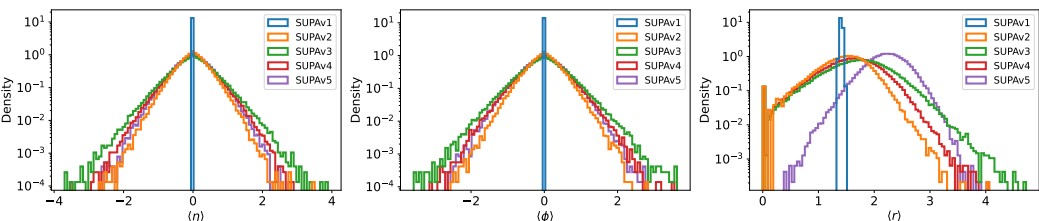

Figure 8: Histograms of feature means

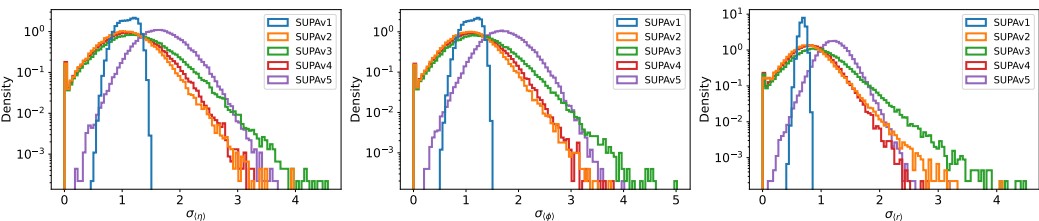

Figure 9: Histograms of feature variances

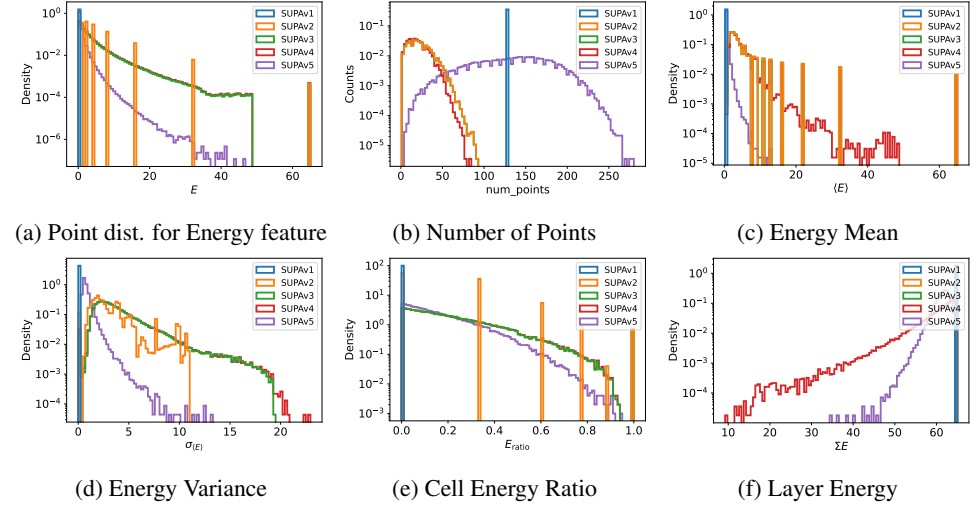

(a) Point dist. for Energy feature     (b) Number of Points     (c) Energy Mean

(d) Energy Variance     (e) Cell Energy Ratio     (f) Layer Energy

Figure 10: Histograms of various shower shape variables

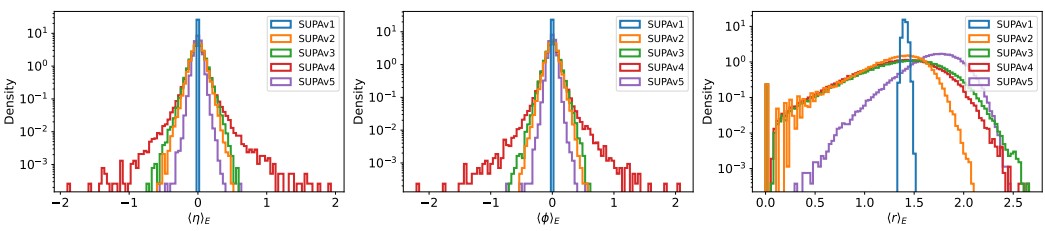

Figure 11: Histograms of layer centroids

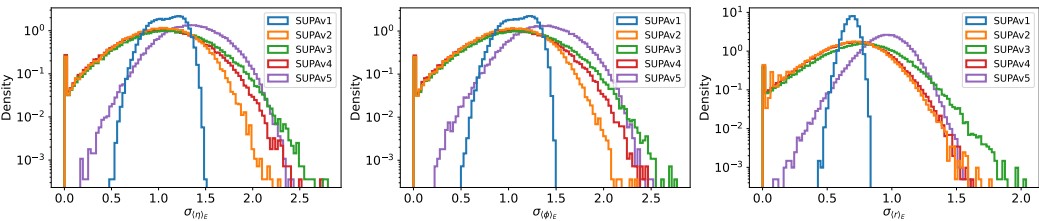

Figure 12: Histograms of layer widths

## A.3 Point Cloud Generative Models

**PointFlow**   PointFlow [Yang et al., 2019] is a flow based model with a PointNet-like encoder and a continuous normalizing flow (CNF) decoder. Additionally, the latents (encoder outputs) are modeled with another CNF to enable sampling. We adapted the PointFlow code to handle variable number of points with masking and masked batch norm. The encoder consists of 1D convolutions with filter sizes $128$, $128$, $256$ and $512$, followed by a three-layer MLP with $256$ and $128$ hidden dimensions to convert the point cloud into its latent representation of size $128$. The CNF decoder has four conditional `concatsquash` layers with a hidden dimension of $128$ and the latent CNF has three `concatsquash` layers with a hidden dimension of $64$. The overall architecture has $0.7M$ trainable parameters.

**SetVAE**   SetVAE Kim et al. [2021] is a transformer-based hierarchical VAE for set-structured data which learns latent variables at multiple scales, capturing coarse-to-fine dependency of the set elements while achieving permutation invariance. We set the number of heads to $4$, the dimension of the initial set to $64$, the hidden dimension to $64$, the number of mixtures for the initial set to $4$, and

the number of inducing points in the hierarchical setup to $[2, 4, 8, 16, 32]$. The overall architecture has $0.5M$ trainable parameters.

**Transflowmer** The *Transflowmer* is flow-architecture using Real NVP layers [Dinh et al., 2016]. As the events are point clouds of varying cardinality, the coupling layers of the flow are required to be permutation equivariant and able to process a varying number of inputs. To satisfy these constraints, we use transformers [Vaswani et al., 2017] without positional encoding in the coupling layers. The overall architecture consists of 16 coupling layers, each of them is parametrised by a 3 transformer layers with $d_{model} = 32$. The overall architecture has $2.1M$ parameters.

We train all the models with $100K$ training examples.

### A.4 Experiments on SUPA datasets

We train point cloud generative models, PointFlow [Yang et al., 2019], SetVAE [Kim et al., 2021], and Transflowmer on SUPA datasets. In this section, we show histogram plots to compare the generative performance across different shower shape variables. For all these plots, the axes limits are chosen according to the ground truth data and generated samples can have probability mass outside the shown range.

#### A.4.1 SUPAv1

Figs. 13 - 18 show the histograms of various shower shape variables for SUPAv1 and samples generated with PointFlow, SetVAE, and Transflowmer.

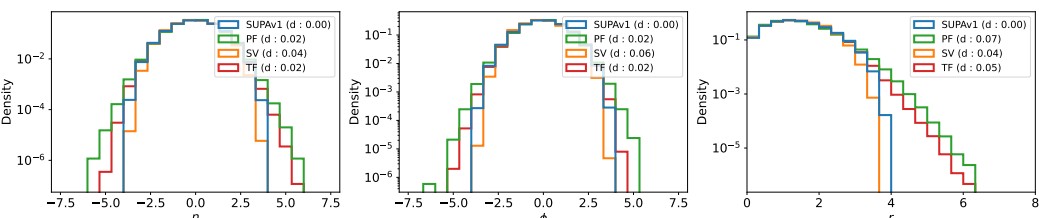

Figure 13: Histograms of point distributions for $\eta$, $\phi$, and $r$

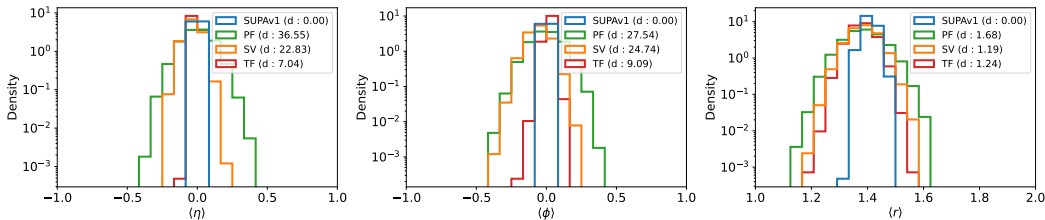

Figure 14: Histograms of sample means for different features

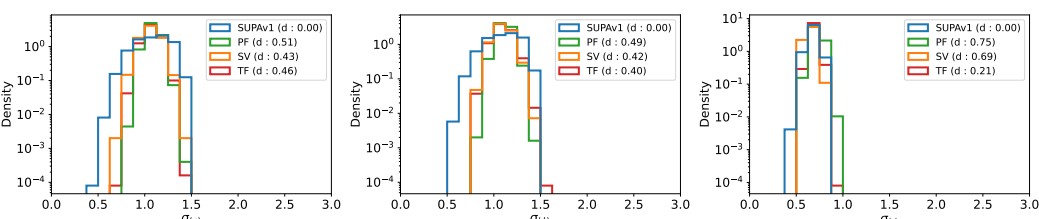

Figure 15: Histograms of sample variance for different features

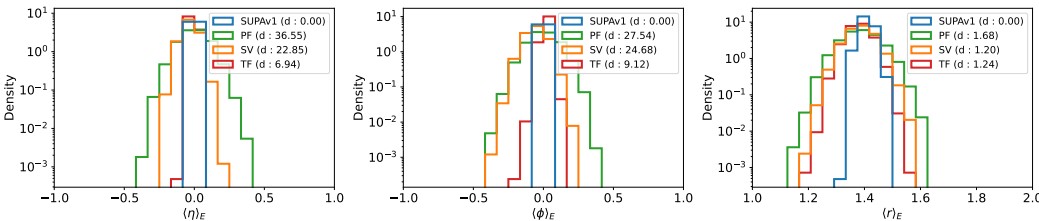

Figure 16: Histograms of energy weighted averages

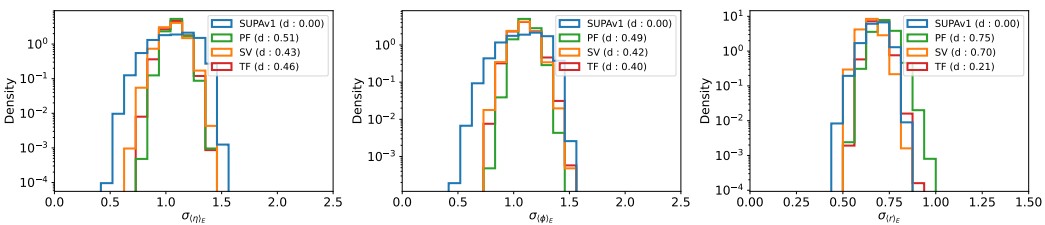

Figure 17: Histograms of lateral widths

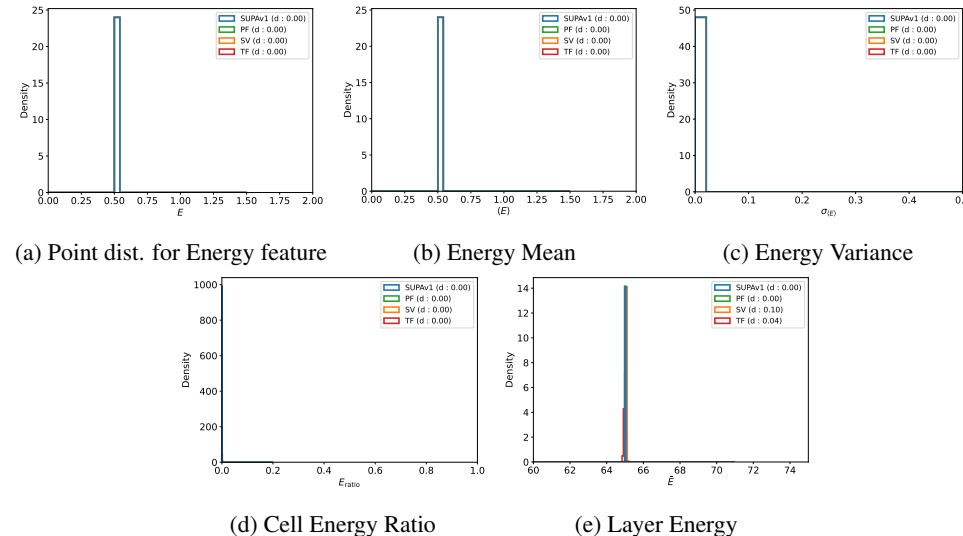

(a) Point dist. for Energy feature     (b) Energy Mean     (c) Energy Variance

(d) Cell Energy Ratio     (e) Layer Energy

Figure 18: Histograms of various shower shape variables

### A.4.2 SUPAv2

Figs. 19 - 24 show the histograms of various shower shape variables for SUPAv2 and samples generated with PointFlow, SetVAE, and Transflowmer.

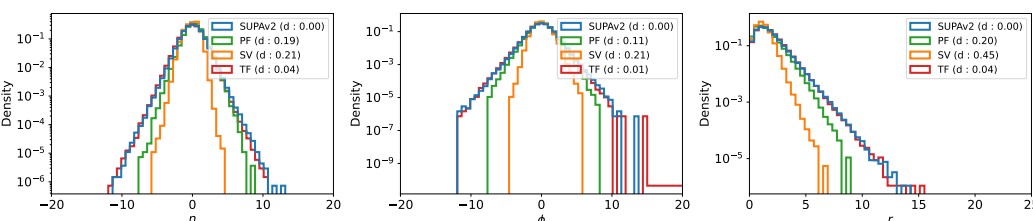

Figure 19: Histograms of point distributions for $\eta$, $\phi$, and $r$

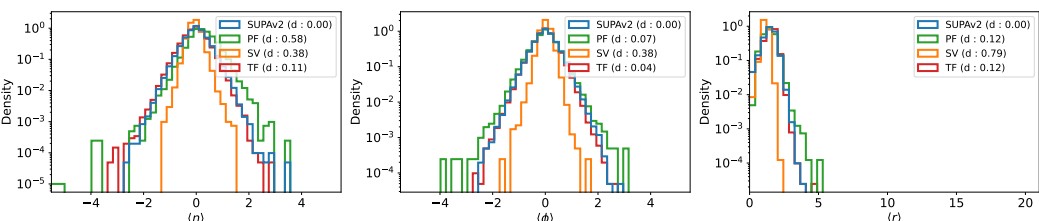

Figure 20: Histograms of sample means for different features

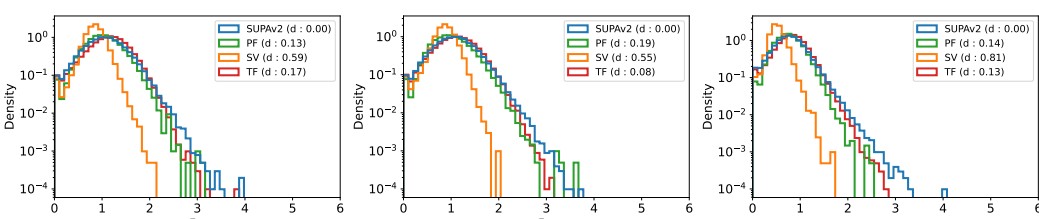

Figure 21: Histograms of sample variance for different features

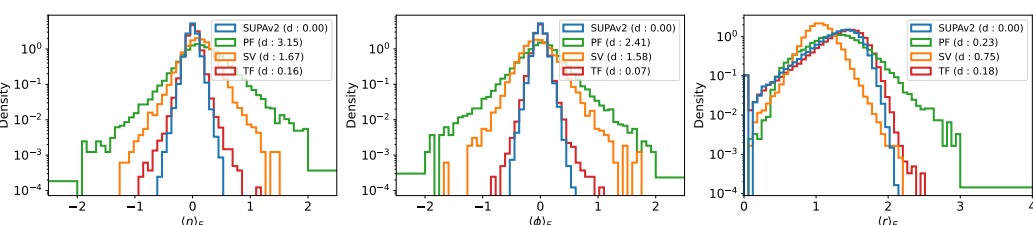

Figure 22: Histograms of energy weighted averages

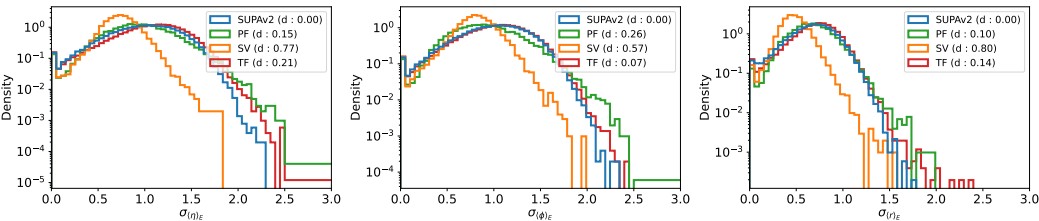

Figure 23: Histograms of lateral widths

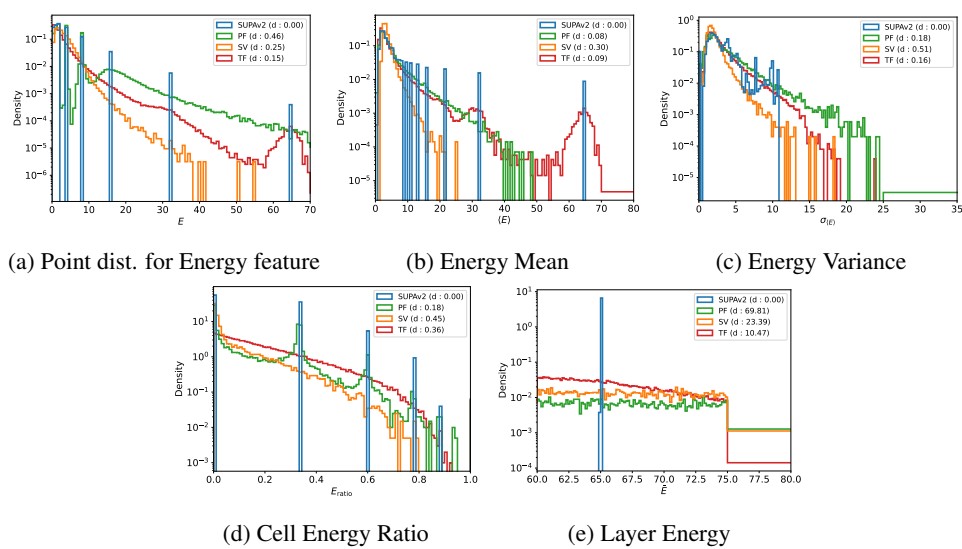

(a) Point dist. for Energy feature  (b) Energy Mean  (c) Energy Variance

(d) Cell Energy Ratio  (e) Layer Energy

Figure 24: Histograms of various shower shape variables

### A.4.3 SUPAv3

Figs. 25 - 30 show the histograms of various shower shape variables for SUPAv3 and samples generated with PointFlow, SetVAE, and Transflowmer.

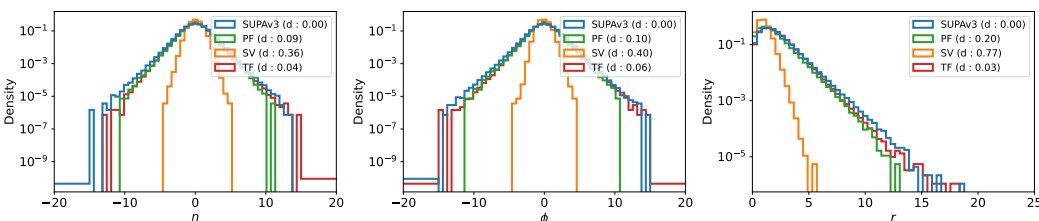

Figure 25: Histograms of point distributions for $\eta$, $\phi$, and $r$

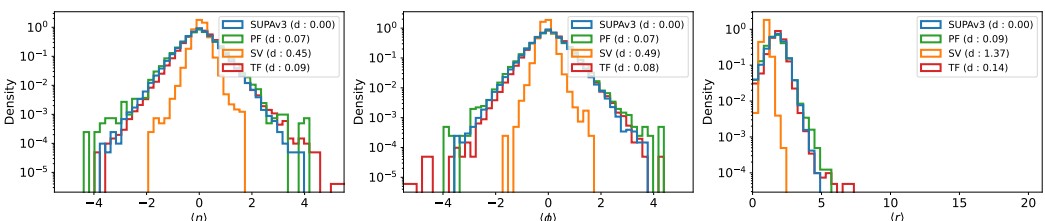

Figure 26: Histograms of sample means for different features

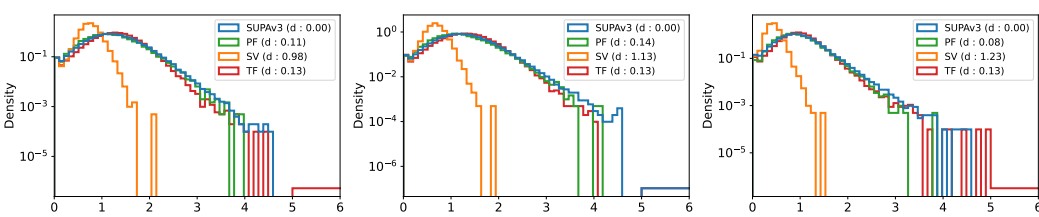

Figure 27: Histograms of sample variance for different features

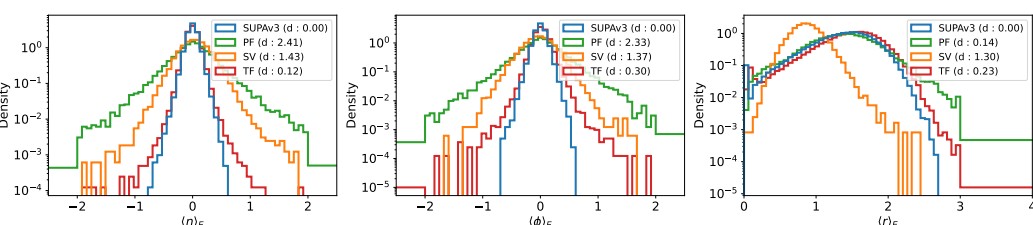

Figure 28: Histograms of energy weighted averages

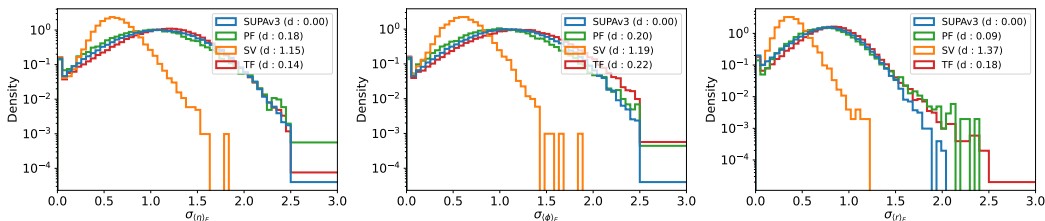

Figure 29: Histograms of lateral widths

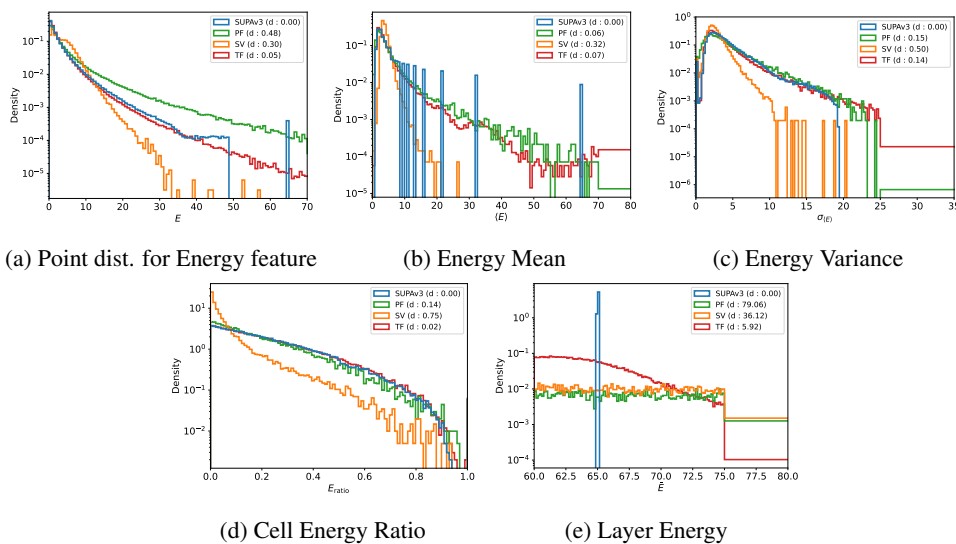

(a) Point dist. for Energy feature      (b) Energy Mean      (c) Energy Variance

(d) Cell Energy Ratio      (e) Layer Energy

Figure 30: Histograms of various shower shape variables

### A.4.4 SUPAv4

Figs. 31 - 36 show the histograms of various shower shape variables for SUPAv4 and samples generated with PointFlow, SetVAE, and Transflowmer.

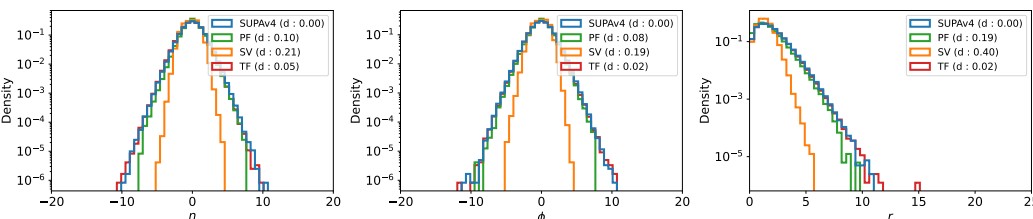

Figure 31: Histograms of point distributions for $\eta$, $\phi$, and $r$

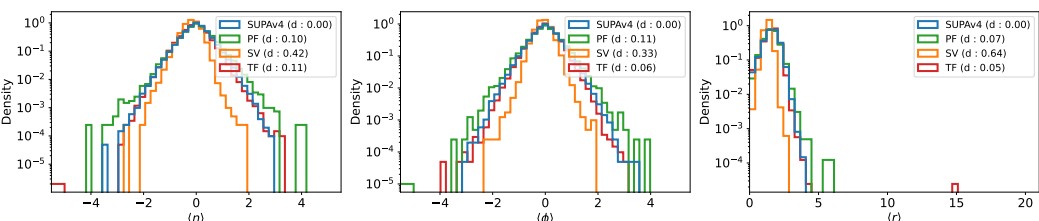

Figure 32: Histograms of sample means for different features

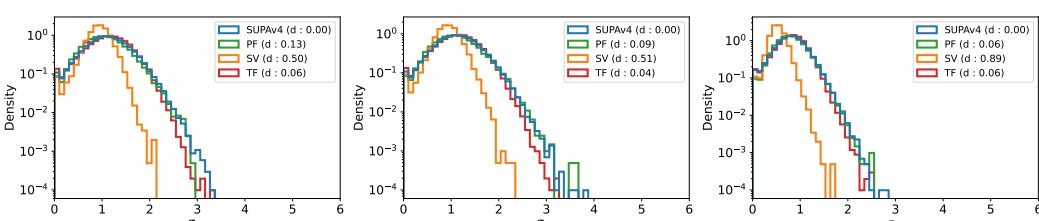

Figure 33: Histograms of sample variance for different features

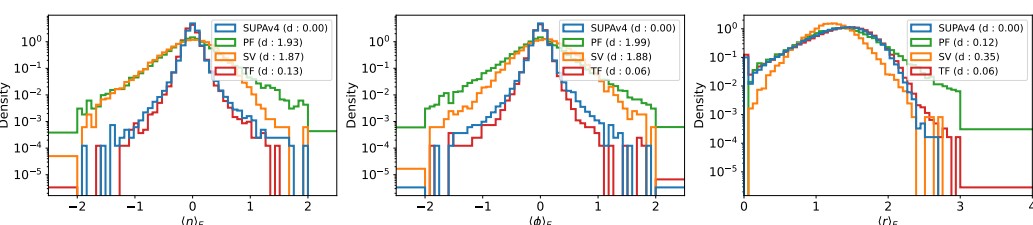

Figure 34: Histograms of energy weighted averages

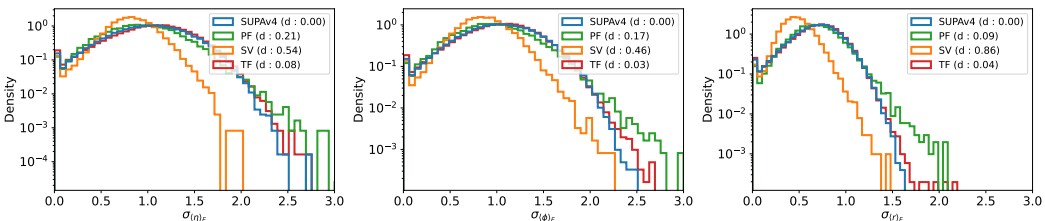

Figure 35: Histograms of lateral widths

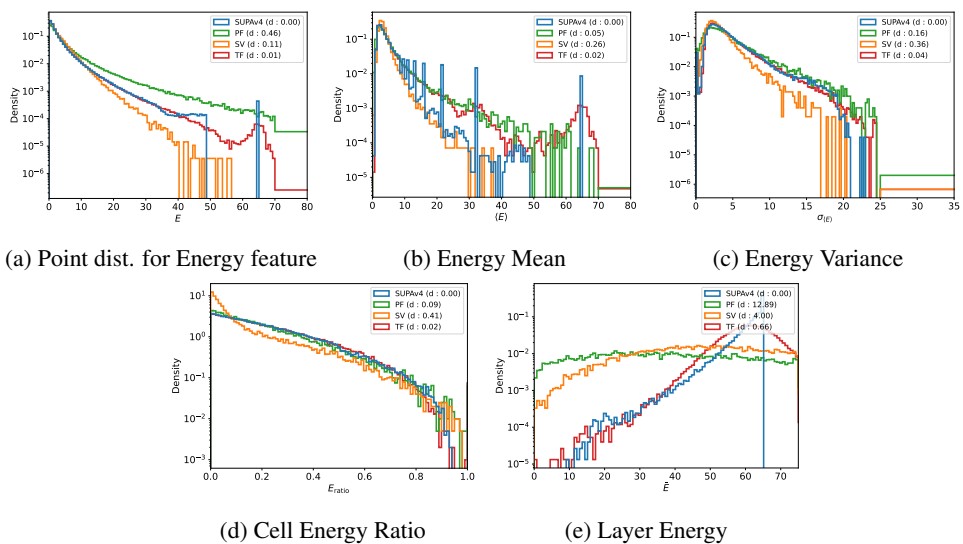

(a) Point dist. for Energy feature      (b) Energy Mean      (c) Energy Variance

(d) Cell Energy Ratio      (e) Layer Energy

Figure 36: Histograms of various shower shape variables

### A.4.5 SUPAv5

We only consider layer 0 for SUPAv5. Figs. 37 - 42 show the histograms of various shower shape variables for SUPAv5 and samples generated with PointFlow, SetVAE, and Transflowmer.

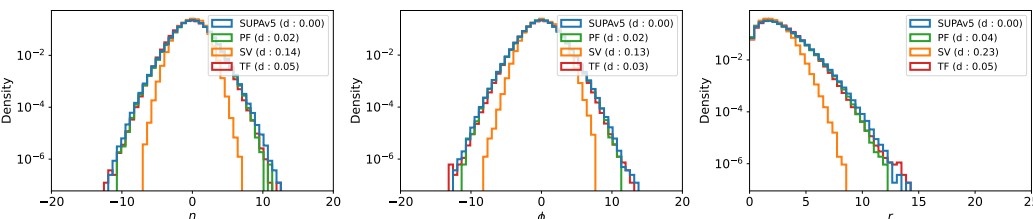

Figure 37: Histograms of point distributions for $\eta$, $\phi$, and $r$

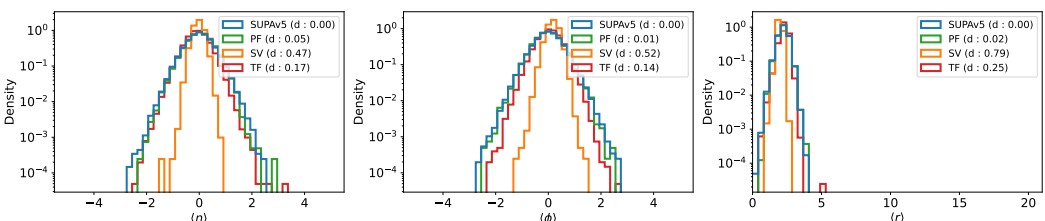

Figure 38: Histograms of sample means for different features

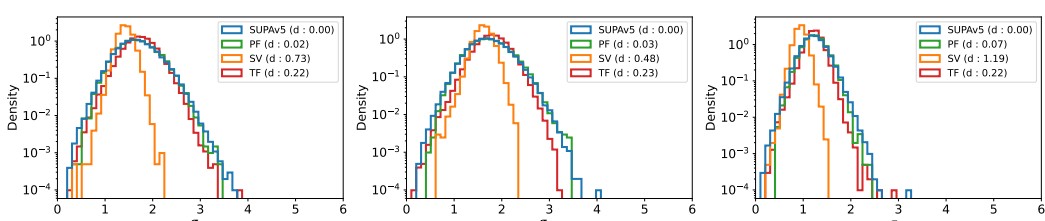

Figure 39: Histograms of sample variance for different features

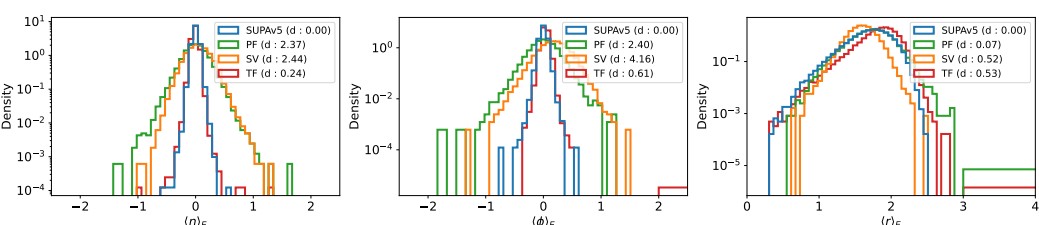

Figure 40: Histograms of energy weighted averages

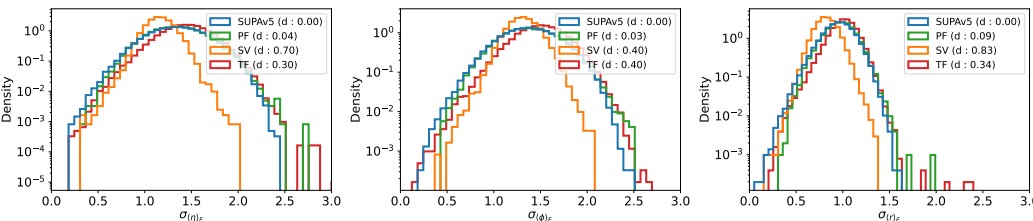

Figure 41: Histograms of lateral widths

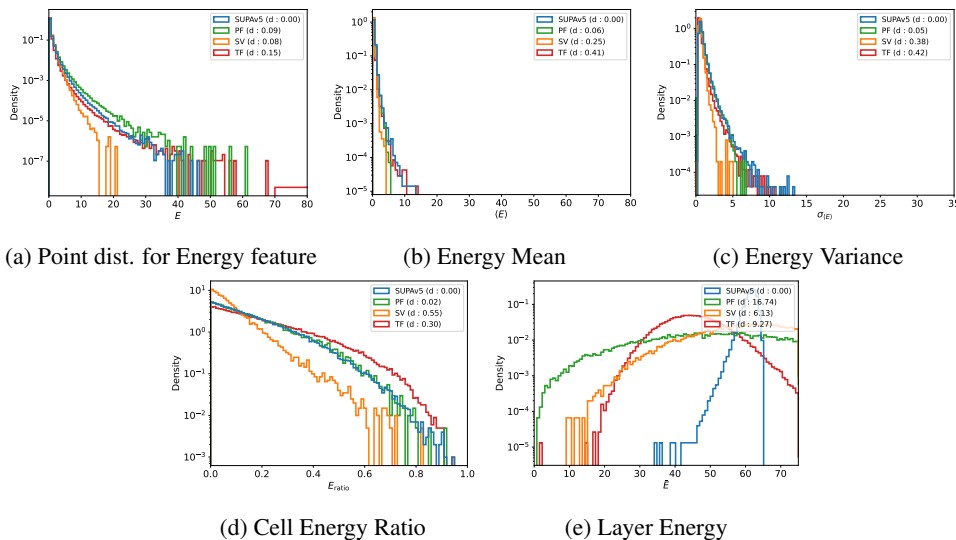

(a) Point dist. for Energy feature    (b) Energy Mean    (c) Energy Variance

(d) Cell Energy Ratio    (e) Layer Energy

Figure 42: Histograms of various shower shape variables

## A.5 Experiments on grid representation of data

In this section we will present some studies on generative modeling with the grid representation of data from SUPA. We discuss about how to downsample the point clouds below. For these studies, we generated another version of the dataset with SUPA such that it is similar to the CALOGAN dataset, i.e., with three layers and downsampled to a resolution in the multiples of $3 \times 96$, $12 \times 12$, and $12 \times 6$, for layer 0, 1, and 2, respectively.

**Downsampling.** For comparison, we downsample the point clouds to their corresponding image representation (see Figure 1) by first defining the region of interest i.e. a rectangular region for each layer and the number of bins/cells/pixels in both the horizontal (or $\eta$) and vertical (or $\phi$) directions. Finally, for each cell, we sum the energy of all the points falling within it to get the pixel intensity. We can increase the number of cells in order to get higher resolutions. Figure 1b, 1c, and 1d show the downsampled image representations at resolutions of 3x, 2x and 1x respectively for the shower shown in Figure 1a. We choose 1x to be the same resolution as used in CaloGAN [Paganini et al., 2018] (i.e. $12 \times 12$ for Layer 1).

### A.5.1 Validity of SUPA as a benchmark with grid representation

We show the comparison of performance of generative models trained over data generated with SUPA and Geant4 in § 5.3. In this section, we extend those studies with more analysis and plots. Figure 43 shows the scatter plot of the average ranks of those models. The average rank for a model on a dataset is obtained by first ranking them with respect to each marginal's discrepancy and then averaging over all the marginals.

Further, in Figures 44-49, we show a subset of the marginals (see § 5 for a detailed explanation on the marginals and Paganini et al. [2018] for the grid representation based marginals) for GEANT4 and

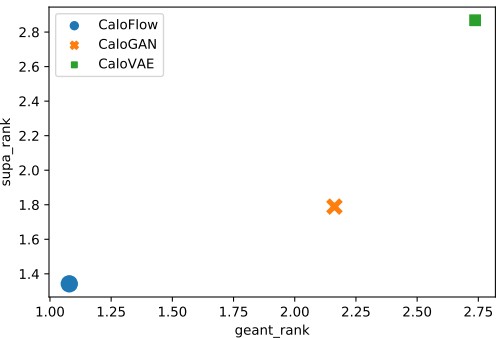

Figure 43: Scatter Plot for ranks over different models. Ranking of the models are consistent over both, SUPA and GEANT4, showing the validity of SUPA as a benchmark.

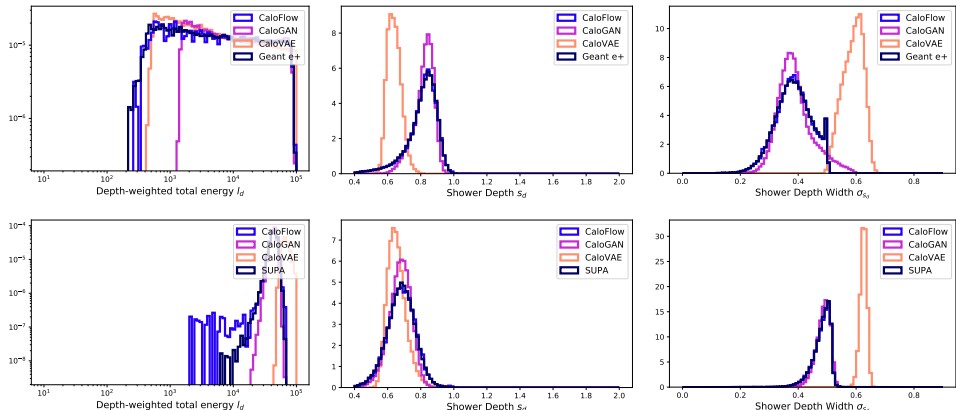

Figure 44: Histogram for various marginals for GEANT4 e+ (top) and SUPA (bottom) vs. showers generated from different trained models

SUPA and also the showers generated with different models trained on them. These marginal plots illustrate the diversity in various distributions present in data from GEANT4, and, more importantly in SUPA. Further, the distributions of the generated showers from different models behave similarly on both datasets, reinstating the proposition that a better model on SUPA implies a better model on the detailed GEANT4.

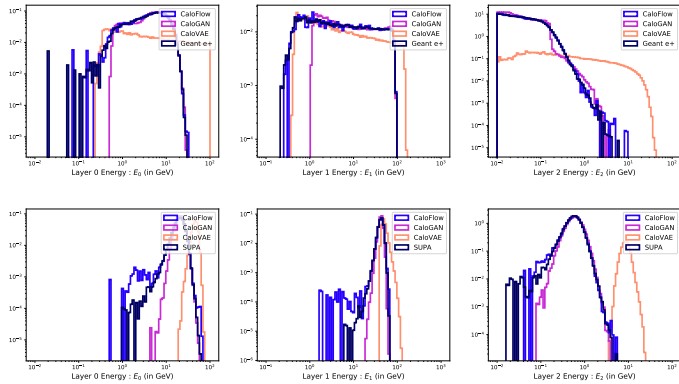

Figure 45: Histogram for Layer Energy for GEANT4 e+ (top) and SUPA (bottom) vs. showers generated with different trained models.

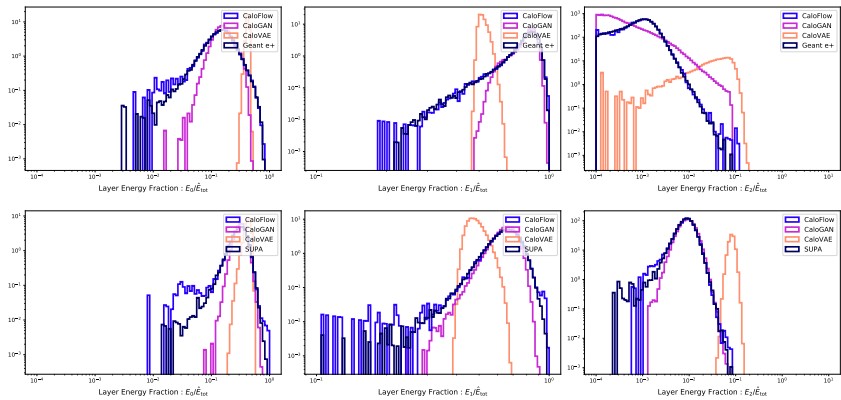

Figure 46: Histogram for Layer energy fraction for GEANT4 e+ (top) and SUPA (bottom) vs. showers generated with different trained models.

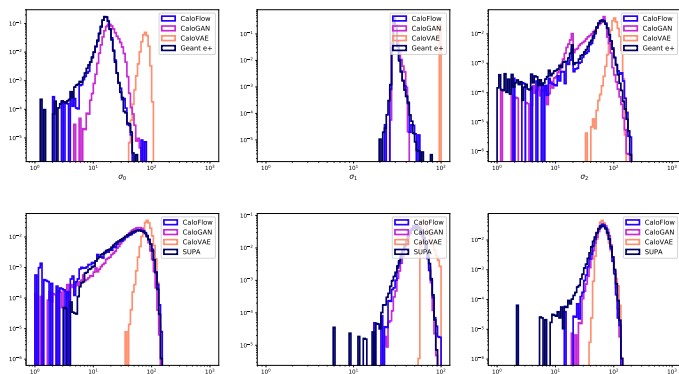

Figure 47: Histogram for Layer lateral width for GEANT4 e+ (top) and SUPA (bottom) vs. showers generated with different trained models.

### A.5.2 High-resolution experiments

In this section, we show the utility of SUPA beyond using it for training at low resolution (similar to the resolution used in CaloGAN, which we call 1x), as well as the limitation of the current models.

|     | 1x   | 2x   | 3x   |
|-----|------|------|------|
| 1x  | 3.57 | 6.35 | 7.20 |
| 2x  | -    | 6.78 | -    |
| 3x  | -    | -    | 8.29 |

Table 4: Mean discrepancy metric (see § A.5.1) for CaloFlow model when trained and tested over different resolutions. Columns correspond to the training resolution and rows to the test resolution. The results on the diagonal show that CaloFlow's performance degrades when resolution increases, and the top row shows that it is not simply due to the sheer dimensionality of the signal since the model does not leverage structure at high resolution to perform better at low resolution.

We train CaloFlow [Krause and Shih, 2021] with SUPA by downsampling the point clouds at the higher resolutions of 2x and 3x. Table 4 shows the mean discrepancy metric (see § A.5.1) for the models. We observe the trend that training at higher resolutions result in poorer performance (diagonal terms) in general. Further, when the generated samples from the trained models are downsampled to 1x, the performance deteriorates as compared to samples generated from models trained directly with data at 1x resolution.

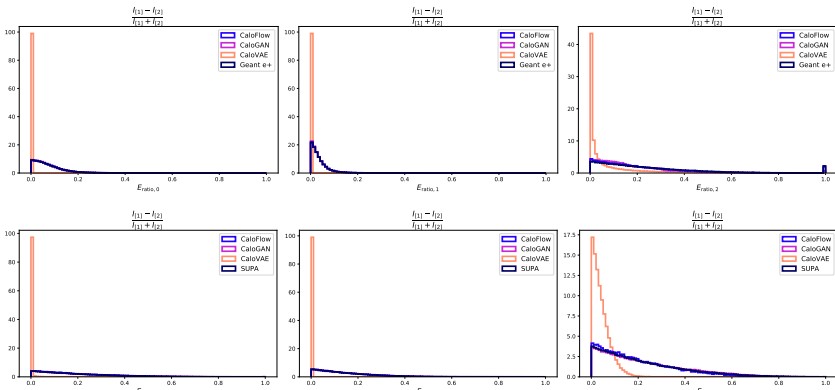

Figure 48: Histogram for $E_{\text{ratio},i}$ for GEANT4 e+ (top) and SUPA (bottom) vs. showers generated with different trained models.

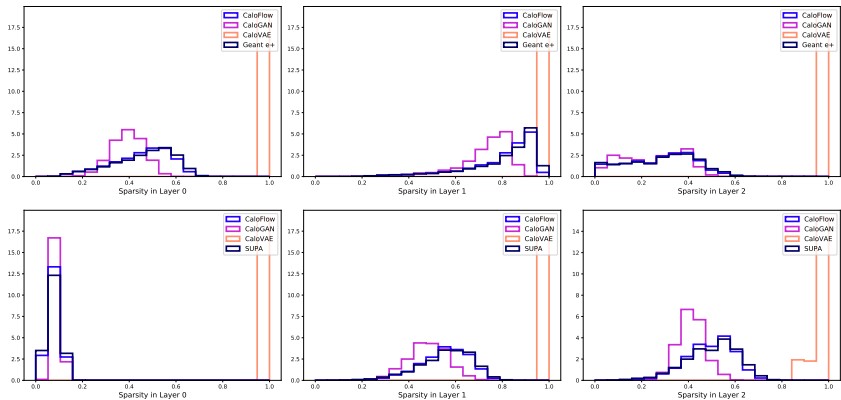

Figure 49: Histogram for Layer sparsity for GEANT4 e+ (top) and SUPA (bottom) vs. showers generated with different trained models.

