# OpenReview forum: "SUPA: A Lightweight Diagnostic Simulator for Machine Learning in Particle Physics"
_NeurIPS.cc/2023/Track/Datasets_and_Benchmarks — NeurIPS 2023 Datasets and Benchmarks Poster_

### Official Review · Reviewer_Y4mj · 2023-07-07
**Machine Learning to Accelerate Particle Showers**

**Rating:** 7
**Confidence:** 3

**Strengths:**

The paper provides a significant advance over earlier work, as it is now feasible to produce benchmark datasets for various configurations, since the ML code runs so much faster than GEANT4. This has the potential to greatly expand what is feasible to test, as much larger and more varied datasets can be produced. So, the broader research community can benefit a lot from the work. As this is generating particle showers that would be produced in particle colliders, there don't seem to be any negative ethical or social implications.

**Additional Feedback:**

None.

**Clarity:**

The paper is very well written. It is well organized and contains the relevant information.

**Correctness:**

The claims appear to be correct. The evaluation and experiments are appropriate and provide meaningful data.

**Documentation:**

The documentation is sufficient. There is sufficient detail to reproduce the results in the paper.

**Ethics:**

No concerns.

**Limitations:**

Yes. The limitations have been adequately described, although some further explanation as described above would be useful.

**Opportunities For Improvement:**

The paper could benefit from further discussion of how comparable the GEANT4 and SUPA output are. In particular, how varying configuration parameters correlates with the output from GEANT4 and how is this confirmed. In addition, it would be helpful to expand the explanation of the results in Table 3. What is the significance of these numbers in terms of the accuracy of the results?

It seems that the appendices were not included in the submission, either as a supplemental pdf or in the main submission, so it isn't possible to assess certain parts of the manuscript in detail. In particular, it would be helpful to have some quantitative information on the statistical comparison between GEANT4 and SUPA output.

**Relation To Prior Work:**

The related work section is very good. There is a clear presentation of prior work and how it relates to the current research, including prior research on using deep learning to generate particle showers.

**Summary And Contributions:**

This paper proposes the use of ML techniques to accelerate the generation of high energy physics particle showers. The resulting code can produce particle showers several orders of magnitude faster than running GEANT4 simulations with comparable properties.

---

### Official Review · Reviewer_q153 · 2023-07-20
**SUPA: A Lightweight Diagnostic Simulator for ML in Particle Physics**

**Rating:** 4
**Confidence:** 4
**Clarity:** Yes

**Strengths:**

1. Paper is clearly written and easy to follow.
2. The simulator is straightforward and Python implementation is helpful for ML practitioners.

**Additional Feedback:**

NA

**Correctness:**

Submission is more of a data generator than a purely constructed, difficult-to-collect benchmark dataset.

**Documentation:**

Yes, the GitHub repo is present with instructions on how to work with the data in a ReadMe file.

**Limitations:**

Major:
1. The paper lacks novelty and compared to the Geant4, their simulator seems incremental contribution.
2. I am concerned this is more of a statistical computing paper where a simulator is more of a focus of this paper than developing an actual benchmark dataset which would be hard to generate for ML practitioners.
3. Appendix is missing from the submission.
4. Benchmark model evaluations are less than extensive.
5. Data size is small. Why not create a more exhaustive benchmark?

Minor:
1. Schematic Fig 2 (left) says vertical lines are slices and layers are a superset of the slices but in line 183 you say layers are a subset of slices.
2. Footnotes 3 and 4 appear before their main text reference.
3. line 196: is N not the total number of slices?

**Opportunities For Improvement:**

NA

**Relation To Prior Work:**

Incremental

**Summary And Contributions:**

The paper introduces a surrogate particle propagation simulator by simulating simplified particle propagation, scattering, and shower development. The proposed simulator generates thousands of particle showers per second which are about 6 orders of magnitudes faster than a high-fidelity simulator named Geant4. The authors present a few point cloud generative baseline models and assess the quality of those generative models for the data generated by their simulator.

---

> ### Author Response · Authors · 2023-08-31
>
> We thank the reviewer for the their valuable feedback and time. We address the weaknesses and questions below and hope that it will lead you to reconsider your recommendation, and/or provide further insights for more improvements.
>
> > The paper lacks novelty and compared  ...
>
> The goal of SUPA is not to replicate Geant4 but to easily simulate similar but simplified processes that still exhibit the same difficulties in training generative models as with Geant4, which we feel arises from the very nature of the data-generating process : stochasticity, scattering, splitting, energy/momentum conservation, etc. We believe SUPA is a computationally light sandbox for benchmarking on tasks (in particular generative modeling) related to calorimetry-like processes, in the same way as MNIST is for natural images (if a model doesn’t work on MNIST, it will most likely not work on natural images, but the reverse cannot be usually pre-determined). Further, SUPA can be tuned to adapt the complexity of generated data. Note that, we do not claim SUPA to be a good pre-training dataset generator.
>
> > I  am concerned this is more of a statistical computing paper ...
>
> SUPA is more of a simple data generator which can be used to generate benchmark datasets. We have improved the code with more comments and added better configuration. As such each config file would correspond to a benchmark if researchers use the generator directly to construct a detector. Further we also propose to release more standardized configs representing a range of complexities.
>
> > Appendix is missing ...
>
> Thanks for pointing it out. We have now added the supplementary material.
>
> > Benchmark model evaluations ...
>
> We have added more plots and comparisons in the appendix.
>
> > Data size is small. Why not create a more exhaustive benchmark?
>
> We will release more standardized configs and corresponding datasets, representing a range of complexities.

---

### Official Review · Reviewer_fTbf · 2023-07-22
**SUPA: A Lightweight Diagnostic Simulator for Machine Learning in Particle Physics**

**Rating:** 9
**Confidence:** 4

**Strengths:**

Because of the lightweight framework for generating datasets, it is very convenient for testing the adaptability of generative models to different detector configurations.  There is a clear value add to the field for SUPA to enable developing more generalizable generative models and the potential for exploring foundation generative models.  The authors should lean into this more by providing datasets that overlap as much as possible with existing work to be a superset — particularly with the 3 Calo Challenge datasets


**Additional Feedback:**

None

**Clarity:**

The paper is clearly written.


**Correctness:**

The dataset generation and models are constructed in a sound way.


**Documentation:**

The code and dataset documentation are minimal but sufficient.  It would be good to make that information (or links to it) available in the main body of the paper


**Limitations:**

A minor comment would be to develop more metrics for testing on SUPA datasets in line with the state of the art of the field


**Opportunities For Improvement:**

It would be nice for configurations to be provided that are as close as possible with the some of the current standard for benchmarks (Calo Challenge).  That way then fast simulation developers could develop models for GEANT based datasets and then test dynamism and adaptability across more of the SUPA datasets.


**Relation To Prior Work:**

Yes

**Summary And Contributions:**

SUPA is a very fast generator of particle interactions and propagation in matter.  It is tunable through a manageable number of parameters.  The authors have baselined 5 different benchmark datasets with 3 different models to assess performance.

---

### Official Review · Reviewer_XfFU · 2023-07-23
**A simple simulator for particle physics**

**Rating:** 5
**Confidence:** 3
**Clarity:** The paper is clearly written.

**Strengths:**

SUPA is 1-4 orders of magnitude faster than Geant-4. It provides a flexible tool to quickly generate toy datasets and test the performance of particle generative models.


**Additional Feedback:**

No additional feedback.

**Correctness:**

I am not an expert in the domain. I cannot evaluate if the SUPA method is correct from a particle physics perspective. However, the benchmark and evaluation methods seem to be correct.


**Documentation:**

The parameters used for dataset generation are clearly documented. Model training, hyperparameters are summarized in the appendix.

**Ethics:**

No ethics concerns.

**Limitations:**

The authors didn’t discuss the negative social impact of their work. I suggest the authors to add a limitation section to discuss the limitation of dataset generated by SUPA compared with Geant-4.

**Opportunities For Improvement:**

- I am not an expert in particle physics, but it seems that SUPA uses a simplified model in algorithm 1 with fixed stopping probability, slitting probability, etc. Some physics is lost compared with Geant-4, e.g. the dynamics of SUPA trajectories may be simpler. It may simplify the task of learning a generative model. I suggest the authors comment on the limitations of the dataset generated by SUPA compared with Geant-4.
- The SUPA code at https://github.com/itsdaniele/SUPA is not very well-documented
- It seems like the dataset generated by SUPA is relatively small and can also be generated by Geant-4 given enough compute. It is unclear whether there is a specific use case where it is infeasible to generate datasets with Geant-4.
- In Figure 4, the NLL of the flow models trained on Geant and SUPA only has qualitative agreement on overall trends.


**Relation To Prior Work:**

The authors only mention Geant-4 as prior work. Are there other widely used software packages for such simulation?

**Summary And Contributions:**

This paper presents SUPA, a python-based simulator to generate data for the particle propagation and shower events in particle physics. The goal is to provide a simple alternative to the widely-used, more expensive simulator, Geant-4, to benchmark machine learning models. It also shows that SUPA gives qualitative similar trends to Geant-4 in terms of model performance.

---

> ### Author Response · Authors · 2023-08-31
>
> We thank the reviewer for the their valuable feedback and time. We address the raised concerns below and hope that it will lead you to reconsider your recommendation, and/or provide further insights for more improvements.
>
> > I am not an expert in particle physics, but it seems that SUPA uses a simplified model ...
>
> Indeed some physics is lost due to simplification and our goal is not to replicate Geant4 but to easily simulate similar but simplified processes that still exhibit the same difficulties in training generative models as with Geant4, which we feel arises from the very nature of the
> data-generating process : stochasticity, scattering, splitting, energy/momentum conservation, etc., and also demonstrate in our analysis (Secs. 5, A.5.). We will add a discussion on limitations of SUPA to make it more clear.
>
> > The SUPA code at https://github.com/itsdaniele/SUPA is not very well-documented
>
> We are consistently enhancing the code's transparency and ease of configuration handling by incorporating additional comments and ongoing refinements.
>
> > It seems like the dataset generated ...
>
> We propose to release more standardized configs representing a range of complexities (including size). As discussed in the introduction, using Geant4 is computationally very expensive, and also requires specific domain knowledge to be set up and tuned.
>
> > In Figure 4, the NLL of ..
>
> We show more plots in the added supplementary material. Please refer to Secs. A.4 and A.5.
>
> > Limitations : The authors didn’t discuss ...
>
> We will be happy to add a discussion on limitations of SUPA as compared to Geant4 in the main paper.

---

### Official Review · Reviewer_hC4c · 2023-08-03
**A simulator for Particle Physics with unclear goals in ML**

**Rating:** 5
**Confidence:** 4

**Strengths:**

- important practical use case with non-standard type of data (particle showers in detectors)
- a parametric model SUPA approximating statistical properties of complex simulators
- generated datasets can be used to pre-train ML models for detection of particles with specific properties

**Additional Feedback:**

- 120:  ... hard. the difficulty ...  -> hard. The difficulty

**Clarity:**

- in general the text is clearly written, however, some important details are missing, see other comments
- it can be good to describe the whole process of the simulator construction in more detail
- the text can be difficult to read for people who are not specialists in particle physics. So it can be reasonable to provide from the very beginning some main ideas of how simulation of particle showers in particle physics is done

**Correctness:**

- It is not clear how the main results of the paper were verified. Did the authors verify their model one some real-world experimental dataset?

**Documentation:**

- Documentation is sufficient

**Ethics:**

- I do not see any ethical concerns

**Limitations:**

- it is not clear how to tune parameters of SUPA to match characteristics of statistical performance of generally recognized simulators. Should we use some Bayesian optimization framework? Or manual try-and-error process? or what?

- The paper is called "A Lightweight Diagnostic Simulator for Machine Learning in Particle Physics". What have we been able to diagnose in particle physics using the proposed simulator? I can not find any specific information in the main text of the paper

- The authors submit their paper in NeurIPS 2023 Track Datasets and Benchmarks. It is not clear how we can use the ideas from the proposed paper as some benchmark. I think, first we should formulate some (possibly, predictive) problem. Then test a number of methods on the generated data and provide some accuracy indicators. And then we will be able to demonstrate the results in greater details and understand which predictive ML method is the best for the considered specific problem

**Opportunities For Improvement:**

- It is not clear whether results of SUPA are adequate in modeling results of more complex simulations.
It is important to propose an approach verifying this adequacy and demonstrate its results. Figures 3 and 4 are not enough

**Relation To Prior Work:**

- Relation to prior work is fully discussed

**Summary And Contributions:**

The authors consider particle physics simulators emulating particle showers in detectors. They proposed some parametric model SUPA to approximate statistical properties of such simulators. They demonstrated that a better model on SUPA implies a better model on a particle simulator and vice versa.

====

After reading the authors' comments and discussions with other reviewers I decided to keep my rating unchanged.
Actually, the paper has some novelty, but still the main issues raised in my review have not been addressed.

---

> ### Author Response · Authors · 2023-08-31
>
> We thank the reviewer for the their valuable feedback and time. We address the main concerns raised below. We hope that it will lead you to reconsider your recommendation, and/or provide further insights for more improvements.
>
> > It is not clear whether results of SUPA ...
>
> The goal of SUPA is not to replicate Geant4 but to easily simulate similar but simplified processes that still exhibit the same difficulties in training generative models as with Geant4, which we feel arises from the very nature of the data-generating process : stochasticity, scattering, splitting, energy/momentum conservation, etc. We believe SUPA is a computationally light sandbox for benchmarking on tasks (in particular generative modeling) related to calorimetry-like processes, in the same way as MNIST is for natural images (if a model doesn’t work on MNIST, it will most likely not work on natural images, but the reverse cannot be usually pre-determined). Further, SUPA can be tuned to adapt the complexity of generated data. Note that, we do not claim SUPA to be a good pre-training dataset generator.
>
> > it is not clear how to tune parameters of SUPA to ...
>
> SUPA provides tunable parameters such as probabilities of splitting and stopping, angular dispersion, etc., as discussed in Sec. A.2.1. These parameters can be used to adapt the complexity of generated data (higher number of splittings, higher dispersion variance, etc. - see Sec. A.2.2 and A.5.1).
>
> > The paper is called "A Lightweight ...
>
> In its current form SUPA can be useful for diagnosing generative models for calorimetery-like data as we have demonstrated in our evaluations. It is also possible to extend SUPA to other tasks such as optimal detector design, etc.
>
> > The authors submit their paper in NeurIPS 2023 Track Datasets and Benchmarks ...
>
> SUPA is more of a simple data generator which can be used to generate benchmark datasets. We have improved the code with more comments and added better configuration. As such each config file would correspond to a benchmark if researchers use the generator directly to construct a detector. Further we also propose to release more standardized configs representing a range of complexities.
>
> > Correctness ...
>
> We have shown these comparisons in the added supplementary in Sec. A.5, as well as Figs. 3 and 4 for which we have used the standard CaloGAN dataset generated with Geant4.
>
> > Clarity ...
>
> We will be happy to add a summary in the introduction regarding detector construction and particle propagation in detectors, and also include related references.

---

### Decision · Program_Chairs · 2023-09-22

**Decision:**

Accept (Poster)

**Comment:**

Paper presents a surrogate model for sampling of particle showers in high-energy partilce detectors, providing realistic QoI data to supplement detailed simulators.
It includes physical bias into the surrogate ansatz.
The paper appears to have potentially high impact within its targeted audience
The reviews indicate that the limitations by approach and in generated datasets could have been addressed more comprehensively.